# Inference-Aware Meta-Alignment of LLMs via Non-Linear GRPO

**Shokichi Takakura** [1]   **Akifumi Wachi** [1]   **Rei Higuchi** [2,3]   **Kohei Miyaguchi** [1]   **Taiji Suzuki** [2,3]

## Abstract

Aligning large language models (LLMs) to diverse human preferences is fundamentally challenging since criteria can often conflict with each other. Inference-time alignment methods have recently gained popularity as they allow LLMs to be aligned to multiple criteria via different alignment algorithms at inference time. However, inference-time alignment is computationally expensive since it often requires multiple forward passes of the base model. In this work, we propose *inference-aware meta-alignment* (IAMA), a novel approach that enables LLMs to be aligned to multiple criteria with limited computational budget at inference time. IAMA trains a base model such that it can be effectively aligned to multiple tasks via different inference-time alignment algorithms. To solve the non-linear optimization problems involved in IAMA, we propose *non-linear GRPO*, which provably converges to the optimal solution in the space of probability measures.

## 1. Introduction

Aligning large language models (LLMs) via reinforcement learning from human feedback (RLHF) (Christiano et al., 2017; Ouyang et al., 2022) or verifiable rewards (RLVR) (Lambert et al., 2024; Shao et al., 2024; Jaech et al., 2024) is a crucial step to ensure that they generate responses aligned with human preferences or specific criteria. However, in real-world applications, the requirements of LLM responses are often diverse depending on the situation and cannot be captured by a single reward function. Several works have explored the idea of aligning LLMs to multiple reward functions simultaneously (Agnihotri et al., 2025; Dai et al., 2024), but it is fundamentally challenging or even impossible since criteria can often conflict with each other (Bai et al., 2022).

Recently, the paradigm of inference-time alignment has gained popularity, where the base model is aligned at inference time using techniques such as best-of-N (BoN) sampling (Nakano et al., 2021), variants of BoN (Verdun et al., 2025; Jinnai et al., 2025), and self-consistency (Wang et al., 2023). Since these inference-time alignment algorithms can modify the outputs of the base model at inference time based on different reward functions or criteria, the model can be aligned to diverse preferences. However, inference-time alignment is computationally expensive since it often requires multiple forward passes of the base model at inference time. With limited computational budget, the performance of inference-time alignment can be poor if the base model is not properly trained. For example, if we aggregate multiple rewards into a single metric and align the model to it, this results in a model that yields mediocre responses across all rewards, leading to a loss of diversity. Consequently, this makes it difficult to sample responses tailored to a specific reward using inference-time alignment methods such as BoN. Thus, we pose the following question: *Can we train a model that can be efficiently aligned to diverse preferences via inference-time alignment?*

To address this issue, we propose *inference-aware meta-alignment* (IAMA), a novel and general framework for aligning LLMs to multiple reward functions via two-stage alignment procedure: 1) *meta-train* the base model with the inference-aware objective and 2) inference-time alignment with different reward functions. We show that the two-stage alignment process can be regarded as a model-agnostic meta-learning (MAML) (Finn et al., 2017) in the space of probability measures and meta-learning problem can be formulated as a non-linear optimization problem with respect to the policy distribution. This is in contrast to the standard expected reward maximization problems, which are linear with respect to the policy distribution. Due to the non-linearity of the objective, we cannot directly apply existing policy optimization algorithms such as TRPO (Schulman et al., 2015), PPO (Schulman et al., 2017), and GRPO (Shao et al., 2024) to solve the problem. To address this issue, we propose a novel algorithm called *non-linear GRPO*, which linearizes the objective at each iteration using the first-order variation. While the optimization landscape of LLMs is generally highly non-convex, our objective in the space of probability measures is shown to be convex for certain

---

[1]LY Corporation [2]The University of Tokyo [3]RIKEN AIP. Correspondence to: Shokichi Takakura <stakakur@lycorp.co.jp>.

*Proceedings of the 43rd International Conference on Machine Learning*, Seoul, South Korea. PMLR 306, 2026. Copyright 2026 by the author(s).

inference-time alignment algorithms such as BoN. Using this property, we prove the linear convergence of the proposed algorithm to the optimal solution. Empirically, we demonstrate that the proposed method can effectively align LLMs to multiple reward functions simultaneously.

Our contribution can be summarized as follows:

- We propose *inference-aware meta-alignment* (IAMA), a novel and general framework which optimizes a single base model so that it can adapt effectively to diverse preferences via inference-time alignment. This can be regarded as a generalization of celebrated MAML framework in the space of probability measures.

- We show that IAMA can be formulated as a non-linear optimization problem in the space of measures and to solve the problem, we develop *non-linear GRPO*, which can be regarded as a mirror descent with respect to KL divergence.

- Theoretically, we prove the convexity of the objective for BoN-type inference-time alignment methods and show that non-linear GRPO converges linearly to the optimal solution. We also empirically demonstrate the effectiveness of the proposed method on multiple tasks and models. While implementation is straightforward, it significantly pushes Pareto frontiers in a multi-objective RLHF task.

## 1.1. Other Related Work

**Inference-aware Alignment**   Several recent works have explored the idea of incorporating inference-time alignment algorithms into the training objective (Chow et al., 2025; Tang et al., 2025; Balashankar et al., 2025). Another line of work has investigated the pass@$k$ or max@$k$ objectives, which is equivalent to applying BoN sampling at inference time (Walder & Karkhanis, 2025; Bagirov et al., 2025). However, these works typically focus on a single inference-time alignment algorithm and a single reward function. Therefore, it is still unclear how to effectively align LLMs to multiple reward functions simultaneously via two-stage alignment. In addition, they consider only BoN sampling and do not provide a general framework that can handle various inference-time alignment algorithms. The only exception is InfAlign framework (Balashankar et al., 2025) but it focuses on a single inference-time alignment algorithm. In addition, while their problem formulation can be applied to general inference-time alignment algorithms, the proposed *InfAlign-CTRL* method is applicable only for BoN-type alignment algorithms and limits the reward functions to be win rate vs. a reference model. Therefore, it is still unclear how to solve the general inference-aware alignment problem even for a single inference-time alignment algorithm. See Appendix B for more discussion.

**Meta-learning**   Meta-learning (Thrun & Pratt, 1998) aims to train models that can quickly adapt to new tasks via few-shot learning or fast adaptation. One of the most popular approaches to meta-learning is MAML (Finn et al., 2017), which learns model parameters such that it can be effectively adapted to multiple tasks via few gradient steps. However, the inference-time alignment procedure does not update the model parameters and thus, cannot be directly handled by the original MAML framework. MetaAlign (Zhang et al., 2025) achieves dynamic alignment by conditioning the model on system prompts that specify preferences. While effective for explicit instruction following, it may struggle to capture complex or implicit preferences that are not easily articulated in prompts.

**Optimization in the Space of Probability Measures**   Optimization problems in the space of probability measures appear in various machine learning problems such as reinforcement learning, variational inference, and generative modeling (Chu et al., 2019). A line of work has explored particle-based methods such as mean-field Langevin dynamics and particle dual averaging (Mei et al., 2018; Nitanda et al., 2021). However, these methods require retraining for each resampling. Another line of work has studied mirror descent (Aubin-Frankowski et al., 2022; Yao et al., 2024) or dual averaging (Kawata et al., 2025) methods in the space of probability measures. Although these methods are theoretically straightforward, they focus on tractable setups, where the first-order variation can be computed in closed form or includes computationally expensive steps such as normalizing flows (Yao et al., 2024). Therefore, it is still unclear how to solve non-linear optimization problems involved in LLM alignment.

## 2. Problem Settings

We consider a language model which is represented as a probability distribution (or policy) $\pi$ over responses $y$ given a context $x$. We denote the space of all possible contexts and responses as $\mathcal{X}$ and $\mathcal{Y}$, respectively. We assume that the contexts are sampled from a distribution $\rho$ over $\mathcal{X}$. We define the space of all possible policies as $\mathcal{P}$ and the reference policy as $\pi_{\text{ref}} \in \mathcal{P}$. Throughout this paper, we often omit the dependence on the context $x$ for simplicity if the notation does not cause confusion. For instance, we write $\pi(y)$ instead of $\pi(y \mid x)$.

**Training-time Alignment**   Let $r : \mathcal{X} \times \mathcal{Y} \to [0, r_{\max}]$ be a reward function which measures the quality of the response $y$ given the context $x$. The objective of alignment is to find a policy $\pi(y \mid x)$ that maximizes the expected reward while staying close to a reference policy $\pi_{\text{ref}}$. Typically, the objective is formulated as a KL-regularized expected reward

maximization problem:

$$\mathcal{R}[\pi] = \mathbb{E}_{y \sim \pi(\cdot|x), x \sim \rho}[r(x, y)] - \beta \mathrm{KL}[\pi \mid \pi_{\mathrm{ref}}],$$

where KL is the Kullback-Leibler divergence defined as $\mathrm{KL}[\pi \mid \pi_{\mathrm{ref}}] = \mathbb{E}_{y \sim \pi(\cdot|x), x \sim \rho}\left[\log \frac{\pi(y|x)}{\pi_{\mathrm{ref}}(y|x)}\right]$ and $\beta \geq 0$ is a regularization parameter.

**Inference-time Alignment** Rarely do we use the aligned model as it is at inference time. Instead, we often employ inference-time alignment algorithms that modify the outputs of the base model $\pi$ at inference time. One of the most popular methods is best-of-N (BoN) sampling (Nakano et al., 2021), where we sample multiple responses from the base model $\pi$ and select the response with the highest reward. Soft BoN (Verdun et al., 2025) is a soft version of BoN, which (approximately) samples responses from an exponentially tilted distribution of rewards. These are equivalent to transforming the base model $\pi$ to a new policy and sampling a response from the transformed policy. We denote such an inference-time alignment algorithm as a functional $\mathcal{T} : \mathcal{P} \to \mathcal{P}$ that maps a base model $\pi \in \mathcal{P}$ to an aligned model $\mathcal{T}[\pi]$. In the case of BoN with $N$ samples, the transformed policy is given as $N \cdot C[\pi](y)^{N-1} \pi(y)$ (Beirami et al., 2025; Gui et al., 2024; Amini et al., 2025) where $C[\pi](y) = \int_{r(y') \leq r(y)} \pi(y') \, dy'$ is the cumulative distribution function of rewards under $\pi$, and in the case of soft BoN with temperature parameter $\tau > 0$, it is given as $\mathcal{T}[\pi](y) \propto \exp(r(y)/\tau)\pi(y)$ (Verdun et al., 2025). Other than (Soft) BoN, there exist various inference-time alignment algorithms such as self-consistency (Wang et al., 2023) and a MCMC-based method (Karan & Du, 2025) and our general framework can handle these algorithms as well.

## 3. Proposed Framework: Inference-aware Meta-Alignment

As we have seen in the previous section, modern alignment procedure often involves two stages: 1) training-time alignment of the base model $\pi$ and 2) inference-time alignment via an algorithm $\mathcal{T}$. Considering the multiple inference-time alignment algorithms $\{\mathcal{T}_i\}_{i=1}^m$ and corresponding reward functions $\{r_i\}_{i=1}^m$, a natural question arises: *What is the best model $\pi$ that can be effectively aligned to diverse rewards via inference-time alignment algorithms?* To address this question, we propose the following objective for training-time alignment:

$$\mathcal{R}[\pi] = \underbrace{g(R_1[\pi], \dots, R_m[\pi])}_{R[\pi]} - \beta \mathrm{KL}[\pi \mid \pi_{\mathrm{ref}}], \quad (1)$$

where $R_i[\pi] = \mathbb{E}_{y \sim \mathcal{T}_i[\pi], x \sim \rho}[r_i(x, y)]$, and $g : \mathbb{R}^m \to \mathbb{R}$ is an aggregation function. The aggregation function $g$ can be weighted sum $\sum_{i=1}^m w_i R_i[\pi]$ for some weights $w_i \geq 0$,

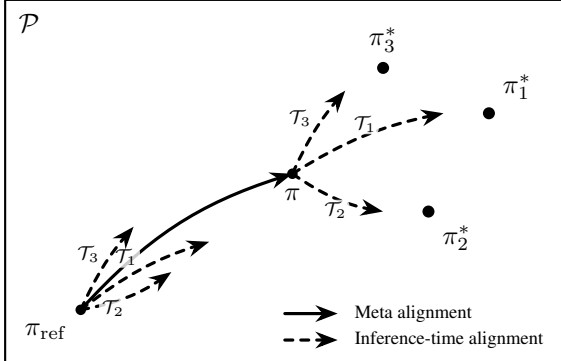

*Figure 1.* Illustration of inference-aware meta-alignment (IAMA). The base model $\pi_{\mathrm{ref}}$ is meta-trained so that it can be effectively adapted to multiple task optima $\{\pi_i^*\}_{i=1}^m$ via inference-time alignment algorithms $\{\mathcal{T}_i\}_{i=1}^m$. Compared to deploying $\pi_{\mathrm{ref}}$ directly, IAMA enables obtaining better aligned models for each task with limited computational budget at inference time.

worst-case performance $\min_i R_i[\pi]$, or smooth minimum $-\frac{1}{\gamma} \log(\sum_{i=1}^m w_i \exp(-\gamma R_i[\pi]))$ for $\gamma > 0$, which interpolates between the average and worst-case performance. This objective can be regarded as a generalization of InfAlign framework (Balashankar et al., 2025) to multiple inference-time alignment algorithms and general reward functions beyond win rate against a reference model.

Similar motivation has been explored in the context of model-agnostic meta-learning (MAML) (Finn et al., 2017), where the goal is to find parameters that can be effectively adapted to multiple tasks via few gradient steps. On the other hand, in the inference-time alignment, the model parameters are not updated during adaptation, but the base model $\pi$ is transformed to a new policy $\mathcal{T}_i[\pi]$ in the space of probability measures. We illustrate the concept of proposed framework in Fig. 1.

While the formulation looks similar to the standard KL-regularized expected reward maximization, the key difference is that the reward term $R[\pi]$ is non-linear with respect to the base model $\pi$ even for a weighted sum due to the presence of the inference-time alignment algorithms $\{\mathcal{T}_i\}_{i=1}^m$. This is in contrast to the expected reward $E_\pi[r]$, which is linear with respect to $\pi$. Due to the non-linearity, we need to develop new algorithms to solve the optimization problem.

### 3.1. Comparison with Naive Approach

A naive approach to align LLMs to multiple reward functions is to train a base model $\pi$ using weighted sum of expected rewards without considering inference-time alignment. That is, we optimize the following objective:

$$\mathcal{R}_{\mathrm{naive}}[\pi] = \sum_{i=1}^m w_i \mathbb{E}_{y \sim \pi, x \sim \rho}[r_i(x, y)] - \beta \mathrm{KL}[\pi \mid \pi_{\mathrm{ref}}].$$

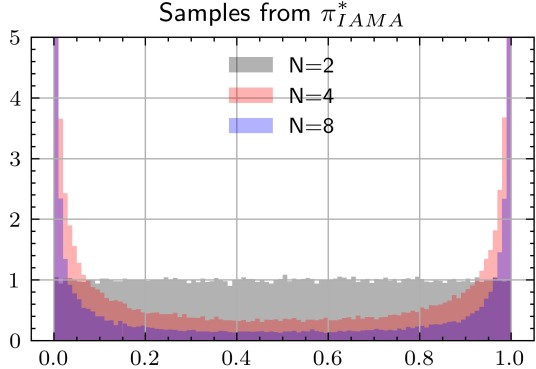

*Figure 2.* Optimal IAMA policies $\pi^*_{\text{IAMA}}$ with BoN sampling ($N = 2, 4, 8$). The optimal policy $\pi^*_{\text{IAMA}}$ with large $N$ produces diverse outputs to cover both modes at $y = 0$ and $y = 1$.

To see the difference between the naive approach and IAMA, let us consider a simple case where $\mathcal{Y} = [0, 1]$. Suppose that there are two reward functions:

$$r_1(y) = 1 - y^2, r_2(y) = 1 - (1 - y)^2.$$

These rewards are completely conflicting since $r_1$ is maximized at $y = 0$ while $r_2$ is maximized at $y = 1$. If we optimize the naive objective with equal weights $w_1 = w_2 = 0.5$ and ignore the KL regularization for simplicity, the optimal policy is given as $\pi^*_{\text{naive}} = \delta_{0.5}$, which always outputs $y = 0.5$. Here, $\delta_a$ is the Dirac delta distribution centered at $a$. On the other hand, if we consider inference-time alignment via BoN sampling with $N \geq 2$, we have the following result:

**Proposition 3.1.** *Consider the above reward functions $r_1$ and $r_2$. Let $\mathcal{T}_1$ and $\mathcal{T}_2$ be BoN sampling with $N \geq 2$. Then, the optimal policy that maximizes the IAMA objective* (1) *with equal weights $w_1 = w_2 = 0.5$ and $\beta = 0$ is given as*

$$\pi^*_{\text{IAMA}}(y) = \frac{\alpha y^{\alpha-1}(1-y)^{\alpha-1}}{(y^\alpha + (1-y)^\alpha)^2}$$

*where $\alpha = \frac{1}{N-1}$.*

See Appendix G.1 for the proof. We show the optimal policy $\pi^*_{\text{IAMA}}$ for $N = 2, 4, 8$ in Fig. 2. In the case of BoN, the base policy generates diverse outputs to cover both modes at $y = 0$ and $y = 1$, which allows the inference-time alignment to select high-reward outputs for both reward functions. This is in contrast to the naive optimal policy $\pi^*_{\text{naive}}$, which produces only mediocre outputs. In addition, the optimal policy varies depending on the computational budget at inference time (i.e., the number of samples $N$). This simple example clearly addresses the question of why IAMA should be considered.

*Remark* 3.2. Another possible approach is to align multiple models $\{\pi_i\}_{i=1}^m$ to each reward function $\{r_i\}_{i=1}^m$ separately

and use one of them at inference time depending on the user's preference (Barrett & Narayanan, 2008; Rame et al., 2023). However, this approach requires storing multiple models, which can be impractical for large models used in real-world applications. In addition, the preference of users may not be known by the provider even at inference time. In such cases, this approach cannot be applied while IAMA can still be used if users select best responses based on their own preferences.

## 4. Proposed Method: Non-linear GRPO

Since the objective in Eq. (1) is non-linear with respect to the policy $\pi$, we cannot directly apply existing policy optimization algorithms such as TRPO (Schulman et al., 2015), PPO (Schulman et al., 2017), and GRPO (Shao et al., 2024). To address this issue, we propose a non-linear GRPO algorithm, which is a non-trivial extension of GRPO-type algorithms (Shao et al., 2024; Zheng et al., 2025; Liu et al., 2025), which sample multiple responses for a given context.

From a different perspective, Balashankar et al. (2025) proposed InfAlign-CTRL algorithm to solve a special case of Eq. (1) using reward transformation but it is applicable only for 1) BoN-type algorithms, 2) win-rate maximization, and 3) single reward setting. This limitation is inevitable since such transformation can be computed only under these constraints. Our approach eliminates all three of these constraints and is therefore of independent interest.

First, we revisit the standard TRPO-type algorithms (including TRPO, PPO, GRPO, and their variants). For notational convenience, we consider the minimization of $\mathcal{L}[\pi] := -\mathcal{R}[\pi]$ instead of maximization of $\mathcal{R}[\pi]$. Let $\pi_t$ be the current policy at iteration $t$. Then, TRPO-type algorithms can be interpreted as an approximation of the following mirror descent method in the space of probability distributions:

$$\pi_{t+1} = \arg\min_\pi \tilde{\mathcal{L}}[\pi] + \frac{1}{\eta}\text{KL}[\pi \mid \pi_t],$$

$$\tilde{\mathcal{L}}[\pi] := -\mathbb{E}_{y \sim \pi_t}\left[\frac{\pi(y)}{\pi_t(y)}r(y)\right] + \beta\text{KL}[\pi \mid \pi_{\text{ref}}],$$

where $\eta > 0$ is a step-size parameter. In TRPO, the proximal term $\text{KL}[\pi \mid \pi_t]$ is replaced with a constraint $\text{KL}[\pi \mid \pi_t] \leq \varepsilon$ for some $\varepsilon > 0$ and in PPO and GRPO, it is replaced with the following clipped surrogate objective.

$$\pi_{t+1} = \arg\min_\pi \tilde{\mathcal{L}}[\pi],$$

$$\tilde{\mathcal{L}}[\pi] := -\mathbb{E}_{y \sim \pi_t}\left[\min\left(\frac{\pi(y)}{\pi_t(y)}r(y), \text{clip}_\epsilon\left(\frac{\pi(y)}{\pi_t(y)}\right)r(y)\right)\right] + \beta\text{KL}[\pi \mid \pi_{\text{ref}}],$$

where $\text{clip}_\epsilon(z) = \max(\min(z, 1+\epsilon), 1-\epsilon)$ for some $\epsilon > 0$.

We extend the above TRPO-type algorithms to handle the non-linear reward term $R[\pi]$. First, we introduce the notion of first-order variation (functional derivative).

**Definition 4.1.** A functional $\frac{\delta R}{\delta \pi} : \mathcal{P} \times \mathcal{Y} \to \mathbb{R}$ is called the first-order variation (or functional derivative) of a functional $R : \mathcal{P} \to \mathbb{R}$ if for all $\pi, \pi' \in \mathcal{P}$,

$$\left. \frac{dR[\pi + \epsilon(\pi' - \pi)]}{d\epsilon} \right|_{\epsilon=0} = \int \frac{\delta R}{\delta \pi}[\pi](y) \cdot (\pi' - \pi)(y) \, dy.$$

Note that the first variation admits a freedom under constant shifts. In the following, we assume that the first variation $\frac{\delta R}{\delta \pi}[\pi](y)$ exists for all $\pi \in \mathcal{P}$ and $y \in \mathcal{Y}$. To simplify the notation, we denote $\frac{\delta R}{\delta \pi}[\pi]$ as $dR[\pi]$ and $\int dR[\pi](y) \cdot g(y) \, dy$ as $\langle dR[\pi], g \rangle$ for a function $g : \mathcal{Y} \to \mathbb{R}$. Using the first-order variation, we approximate the objective $R[\pi]$ around the current policy $\pi_t$ as

$$R[\pi] \approx R[\pi_t] + \int \frac{\partial R}{\partial \pi}[\pi_t](y) \cdot (\pi(y) - \pi_t(y)) \, dy.$$

Using this approximation, we define the surrogate loss as

$$\tilde{\mathcal{L}}[\pi] = - \left( R[\pi_t] + \int \frac{\partial R}{\partial \pi}[\pi_t](y) \cdot (\pi(y) - \pi_t(y)) \, dy \right)$$
$$+ \beta \mathrm{KL}[\pi \mid \pi_{\mathrm{ref}}]$$
$$= -\mathbb{E}_{y \sim \pi_t} \left[ \frac{\pi}{\pi_t} \frac{\partial R}{\partial \pi}[\pi_t](y) \right] + \beta \mathrm{KL}[\pi \mid \pi_{\mathrm{ref}}] + \mathrm{const}.$$

Then, we update the policy by solving the following proximal objective:

$$\pi_{t+1} = \underset{\pi}{\arg\min} \, \tilde{\mathcal{L}}[\pi] + \frac{1}{\eta} \mathrm{KL}[\pi \mid \pi_t].$$

Aside from the fact that the term $\mathrm{KL}[\pi \mid \pi_{\mathrm{ref}}]$ remains the same, the above update can be interpreted as a mirror descent in the space of probability measures with KL divergence as the Bregman divergence.

While the above update is conceptually straightforward, it is not directly implementable in practice since the first-order variation $\frac{\partial R}{\partial \pi}[\pi_t](y)$ generally depends on the current policy $\pi_t$ in a complex manner. To overcome this issue, we propose to approximate it using empirical samples. In GRPO-type algorithms, we sample multiple responses from the current policy $\pi_t$ for a given context $x$. We denote the sampled responses as $\{y_j\}_{j=1}^{M}$ and the empirical distribution of the samples as $\hat{\pi}_t = \frac{1}{M} \sum_{j=1}^{M} \delta_{y_j}$. Note that we can use other approximations such as kernel density estimation to obtain a smoother approximation of the current policy (Silverman, 2018). Then, we approximate the first-order variation as $\frac{\partial R}{\partial \pi}[\hat{\pi}_t](y)$. Using this approximation, we define the surrogate loss as

$$\hat{\mathcal{L}}[\pi] = -\mathbb{E}_{y \sim \pi_t} \left[ \frac{\pi}{\pi_t} \frac{\partial R}{\partial \pi}[\hat{\pi}_t](y) \right] + \beta \mathrm{KL}[\pi \mid \pi_{\mathrm{ref}}],$$

and we update the policy by approximately solving the following proximal objective:

$$\pi_{t+1} \simeq \underset{\pi}{\arg\min} \, \hat{\mathcal{L}}[\pi] + \frac{1}{\eta} \mathrm{KL}[\pi \mid \pi_t].$$

The above update can be interpreted as a GRPO update with a modified reward function $\tilde{r}_t(y) = \frac{\partial R}{\partial \pi}[\hat{\pi}_t](y)$. That is, we can modify the original GRPO by only introducing a drop-in reward function while keeping the rest of the algorithm unchanged. This makes the non-linear GRPO algorithm easy to implement in practice. Note that we can also use other GRPO-type algorithms such as GSPO (Zheng et al., 2025) as the base algorithm instead of GRPO. In addition, since non-linear GRPO is quite general, it can be applied to various problems beyond IAMA and includes some existing algorithms as special cases. See Appendix C for more discussion. We summarize the non-linear GRPO algorithm in Algorithm 1. In practice, we can further approximate the above update by replacing the expectation with empirical samples and proximal term with the clipped surrogate objective as in PPO and GRPO. Please refer to Algorithm 2 for the full description of practical implementation of non-linear GRPO.

---

**Algorithm 1** Non-linear GRPO

---

1: **Input:** Initial policy $\pi_0$, reference policy $\pi_{\mathrm{ref}}$, objective $R$, regularization parameter $\beta$, step size $\eta$, number of iterations $T$
2: **for** $t = 0, 1, \ldots, T - 1$ **do**
3:     Sample $M$ responses $\{y_j\}_{j=1}^{M}$ from $\pi_t$
4:     Compute approximated functional derivative: $\tilde{r}_t(y) = \frac{\partial R}{\partial \pi}[\hat{\pi}_t](y)$.
5:     Define surrogate loss: $\hat{\mathcal{L}}[\pi] = -E_{y \sim \pi_t}[\frac{\pi(y)}{\pi_t(y)} \cdot \tilde{r}_t(y)] + \beta \mathrm{KL}[\pi \mid \pi_{\mathrm{ref}}]$
6:     Update policy: $\pi_{t+1} \simeq \arg\min_{\pi} \hat{\mathcal{L}}[\pi] + \frac{1}{\eta} \mathrm{KL}[\pi \mid \pi_t]$
7: **end for**
8: **Return:** Final policy $\pi_T$

---

### 4.1. First-Order Variation for (Soft) BoN

Since our formulation is quite general, it can accommodate various inference-time alignment algorithms. In this section, we consider two important cases including 1) BoN and 2) soft BoN (exponential tilting) and derive the concrete forms of the functional derivatives required to implement the non-linear GRPO algorithm. From the chain rule, the first variation of $R$ can be computed as

$$\frac{\partial R}{\partial \pi}[\pi](y) = \sum_{i=1}^{m} \frac{\partial g}{\partial R_i}(R_1[\pi], \ldots, R_m[\pi]) \cdot \frac{\partial R_i}{\partial \pi}[\pi](y).$$

Thus, to compute the first variation of $R$, it suffices to compute the first-order variations of $R_i$ for each inference-time

alignment algorithm $\mathcal{T}_i$ and reward function $r_i$. The following proposition provides formulas for the first-order variations of $R_i$ for BoN and soft BoN, respectively.

**Proposition 4.2.** *Let $R[\pi] = \mathbb{E}_{y \sim \mathrm{BoN}_N[\pi]}[r(y)]$. Assume that the distribution of $r(y)$ under $\pi$ has a density. Then, the first-order variation of $R$ can be computed as*

$$\frac{\partial R}{\partial \pi}[\pi](y) = -\int_{r(y)}^{r_{\max}} N \cdot (C[\pi](r))^{N-1} \, \mathrm{d}r.$$

**Proposition 4.3.** *Let $R[\pi] = \mathbb{E}_{y \sim \mathrm{SoftBoN}_\tau[\pi]}[r(y)]$. Then, the first-order variation of $R$ can be computed as*

$$\frac{\partial R}{\partial \pi}[\pi](y) = \frac{r(y)\exp(r(y)/\tau)}{\int \exp(r(z)/\tau) \cdot \pi(z)\,\mathrm{d}z}$$
$$- \frac{\exp(r(y)/\tau) \cdot \int r(z)\exp(r(z)/\tau) \cdot \pi(z)\,\mathrm{d}z}{\left(\int \exp(r(z)/\tau) \cdot \pi(z)\,\mathrm{d}z\right)^2}.$$

See Appendix F for the proofs. The above formulas involve integrals but they are reduced to sum over empirical samples if we replace $\pi$ with the empirical distribution $\hat{\pi}$. See Appendix D.1 for details. Therefore, we can easily compute the approximated functional derivatives required in the non-linear GRPO algorithm.

# 5. Convergence Analysis

In this section, we provide the convergence guarantees of the non-linear GRPO algorithm. The optimization of LLMs in the space of parameters is generally non-convex and challenging to analyze. On the other hand, if we regard the optimization in the space of probability distributions (policies), the problem becomes simpler. Specifically, in the case of weighted sum and (smooth) minimum of BoN-type objectives, $R[\pi]$ is concave as shown in Section 5.3. We also discuss other cases where $R[\pi]$ is concave in Appendix C.

First, we provide convergence guarantees of non-linear GRPO for general smooth and concave objectives. Note that the concavity is required only for the convergence analysis and the proposed algorithm can be applied to non-concave objectives (e.g. Soft BoN, Worst-of-N (Balashankar et al., 2025) case) as well. Here, we assume the following about the objective function:

**Assumption 5.1.** *The reward functional $R$ is concave and $L$-smooth relative to the KL-divergence. That is, for all $\pi, \pi' \in \mathcal{P}$, we have*

$$R[\pi'] \le R[\pi] + \langle \mathrm{d}R[\pi], \pi' - \pi \rangle,$$
$$R[\pi'] \ge R[\pi] + \langle \mathrm{d}R[\pi], \pi' - \pi \rangle - L \cdot \mathrm{KL}[\pi' \mid \pi].$$

*for some $L > 0$.*

Since $R[\pi]$ is concave, the objective $\mathcal{L}[\pi] = -R[\pi] + \beta\mathrm{KL}[\pi \mid \pi_{\mathrm{ref}}]$ is strongly convex with respect to the KL-

divergence if $\beta > 0$. Therefore, the optimization problem has a unique optimal policy $\pi_* \in \mathcal{P}$.

## 5.1. Convergence Guarantees with Exact Proximal Updates

In this section, we assume that the proximal objective is solved exactly at each iteration. That is, at step $t$, $\pi_{t+1}$ can be computed as

$$\pi_{t+1} = \operatorname*{argmin}_\pi \tilde{\mathcal{L}}[\pi] + \frac{1}{\eta}\mathrm{KL}[\pi \mid \pi_t].$$

Then, we have the following convergence guarantee:

**Theorem 5.2.** *Let $\{\pi_t\}_{t=0}^T$ be the sequence of policies generated by the (exact) non-linear GRPO algorithm with step size $\eta = 1/L$. Then, we have*

$$\mathcal{L}[\pi_T] - \mathcal{L}[\pi_*] \le \frac{\beta \cdot \mathrm{KL}[\pi_* \mid \pi_0]}{((L+\beta)/L)^T - 1}.$$

See Appendix H.1 for the proof. The proof strategy basically follows that of Aubin-Frankowski et al. (2022) except for some modifications to handle KL-regularization. This theorem shows that the non-linear GRPO algorithm converges to the optimal policy exponentially fast if $\beta > 0$.

## 5.2. Convergence Guarantees with Inexact Proximal Updates

Next, we consider the practical case, where the first-order variation is approximated using empirical samples and the proximal objective is solved approximately. That is, at step $t$, we compute $\pi_{t+1}$ as an approximate solution to

$$\pi_{t+1} \approx \operatorname*{argmin}_\pi \hat{\mathcal{L}}[\pi] + \frac{1}{\eta}\mathrm{KL}[\pi \mid \pi_t].$$

Note that $\frac{\partial R}{\partial \pi}[\hat{\pi}_t]$ is a random variable depending on the sampled responses $\{y_j\}_{j=1}^M$. The first-order optimality condition of proximal objective gives $\frac{\partial \hat{\mathcal{L}}}{\partial \pi}[\pi] + \frac{1}{\eta}\log\frac{\pi}{\pi_t} = \mathrm{const}$. In practice, due to the limited representation ability of the model and inexact optimization, there exists a residual term $r_t := \frac{\delta \hat{\mathcal{L}}}{\delta \pi}[\pi_{t+1}] + \frac{1}{\eta}\log\frac{\pi_{t+1}}{\pi_t}$. Then, we have the following convergence guarantee:

**Theorem 5.3.** *Assume that the approximated first-order variation satisfies $\mathbb{E}\left[\left\|\frac{\partial R}{\partial \pi}[\hat{\pi}_t](y) - \frac{\partial R}{\partial \pi}[\pi_t](y)\right\|_{\mathrm{sp}}^2\right] \le \varepsilon$ for some $\varepsilon \ge 0$, and the optimization residual satisfies $\mathbb{E}\left[\|r_t\|_{\mathrm{sp}}^2\right] \le \delta$ for some $\delta \ge 0$ given $\pi_t$. Here, $\|f\|_{\mathrm{sp}} = (\sup_y f(y) - \inf_y f(y))/2$ is the span seminorm. Let $\{\pi_t\}_{t=0}^T$ be the sequence of policies generated by the (inexact) non-linear GRPO algorithm with step size $\eta = 1/L$ and a random index $\hat{t} \in \{1, \ldots, T\}$ following the distribu-*

*tion:* $\text{Prob}\left(\hat{t} = t\right) \propto \left(\frac{L+\beta/2}{L}\right)^{t}$. *Then, we have*

$$\mathbb{E}\left[\mathcal{L}[\pi_{\hat{t}}] - \mathcal{L}[\pi_*]\right] \leq \frac{(\beta/2) \cdot \text{KL}[\pi_* \mid \pi_0]}{((L + \beta/2)/L)^T - 1} + \frac{2(\varepsilon + \delta)}{\beta}.$$

*Here, the expectation is taken over the randomness in the approximation of the functional derivative, optimization procedure, and the choice of $\hat{t}$.*

See Appendix H.2 for the proof. Even with inexact proximal updates, the non-linear GRPO algorithm converges to a neighborhood of the optimal policy at an exponential rate but the bias term appears due to the approximation error in the functional derivative.

In general, the required number of samples $M$ to control the approximation error $\varepsilon$ can be exponentially large in the dimension of the response space $\mathcal{Y}$. However, in most cases, measure transformation $\mathcal{T}$ depends on the distribution of the reward $r(y)$ rather than the response $y$ itself. Thus, since $r(y)$ is one-dimensional, we can expect that the approximation error $\varepsilon$ can be controlled with a moderate number of samples. We provide a dimension-independent bound on the approximation error for BoN-type objectives in the next section.

### 5.3. Properties of BoN-type Objectives

In this section, we prove the concavity and smoothness of $R[\pi]$ for BoN-type alignment algorithms. Then, we derive a dimension-independent bound on the approximation error in the functional derivative. Specifically, we consider the following general form:

$$\mathcal{T}[\pi](y) = f(C[\pi](r(y))) \cdot \pi(y),$$

where $f : \mathbb{R} \to \mathbb{R}$ is monotonically increasing and $L_f$-Lipschitz continuous in $[0, 1]$. This form includes BoN with $f(z) = N \cdot z^{N-1}$ and Best of Poisson (BoP) (Khalaf et al., 2025) (drawing the number of samples $N$ from a Poisson distribution with parameter $\lambda > 0$) with $f(z) = \lambda \cdot e^{\lambda(z-1)}$ as special cases.

**Concavity and Smoothness** For BoN-type algorithms, we have the following results:

**Lemma 5.4.** $R[\pi] = \int r(y)T[\pi](y)\,\mathrm{d}y$ *is concave in $\pi$.*

**Lemma 5.5.** $R[\pi] = \int r(y)\mathcal{T}[\pi](y)\,\mathrm{d}y$ *is $L_f \cdot r_{\max}$-smooth relative to the KL-divergence.*

See Appendix H.3 and H.4 for the proof, respectively. These lemmas show that BoN-type objectives satisfy the assumptions required for the convergence analysis. Note that the concavity is preserved even if we consider the weighted sum or (smooth) min aggregation of multiple BoN-type objectives.

**Approximation Error** Finally, we analyze the approximation error in the functional derivative for BoN-type objectives. The following lemma bounds the approximation error in the functional derivative when using empirical samples.

**Lemma 5.6.** *Let $\{y_j\}_{j=1}^{M}$ be i.i.d. samples from $\pi_t$ and $\hat{\pi}_t = \frac{1}{M}\sum_{j=1}^{M}\delta_{y_j}$ be the empirical distribution. Then, the approximation error in the functional derivative can be bounded as*

$$\mathbb{E}\left[\left\|\frac{\partial R}{\partial \pi}[\hat{\pi}_t](y) - \frac{\partial R}{\partial \pi}[\pi_t](y)\right\|_{\text{sp}}^2\right] \leq \frac{L_f^2 \cdot r_{\max}^2}{M}.$$

See Appendix H.5 for the proof. This lemma shows that the approximation error decreases at the rate of $O(1/M)$ and is independent of the dimension of the response space $\mathcal{Y}$. We also demonstrate in the numerical experiments that the non-linear GRPO algorithm works well with a moderate number of samples $M$.

## 6. Numerical Experiments

In this section, we empirically evaluate the effectiveness of the proposed framework and algorithm through extensive experiments on multiple models and tasks. We use TRL (von Werra et al., 2020) library to implement (non-linear) GRPO algorithm. As discussed in the previous sections, the non-linear GRPO can be implemented by simply replacing the reward function with the approximated functional derivative. Throughout the experiments, we set the number of sampled responses $M = 8$, which is a default value in TRL library. See Appendix I for the detailed experimental settings and Appendix J for additional experimental results, including experiments with different hyperparameters and base models, as well as an analysis of the effect of sample size $M$. We also provide the implementation code in the supplementary material for reproducibility.

### 6.1. Length Reward

We begin with a simple task using length reward to illustrate the difference between aligned policy and meta-aligned policy and see that the proposed method can obtain IAMA solution effectively. Specifically, we consider the following two rewards:

$$r_1(y) = -\left(\frac{|y - L_1|}{L_{\max}}\right)^2, r_2(y) = -\left(\frac{|y - L_2|}{L_{\max}}\right)^2,$$

where $L_1 = 50, L_2 = 150$ are the target lengths and $L_{\max} = 256$ is the maximum length. These reward functions encourage the model to generate responses with lengths close to $L_1$ and $L_2$, respectively. This setup corresponds to the situation where some users prefer short responses while others prefer long and detailed ones. This is

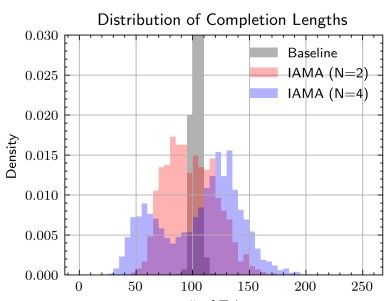 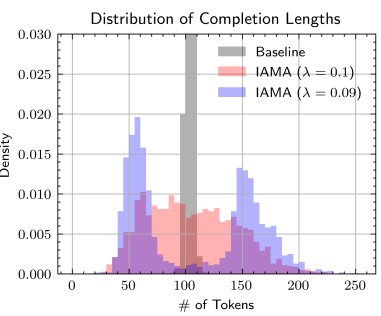 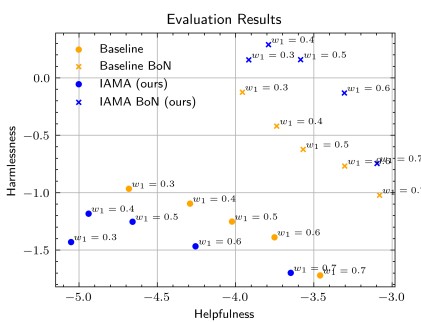

*(a)* Response length distributions for different alignment methods on the length reward task. The meta-aligned model via non-linear GRPO can effectively adapt to both short and long response preferences via (Soft) BoN sampling, while the aligned model via standard GRPO generates only medium-length responses.

*(b)* Averaged rewards on the evaluation set. The meta-aligned model via non-linear GRPO can effectively trade off between helpfulness and harmlessness via BoN ($N = 4$) sampling.

a similar setting to the one in Proposition 3.1. We use Ultra-Feedback dataset (Cui et al., 2023) as the context data and train Mistral 7B Instruct (Jiang et al., 2023) model to optimize the objective $\mathcal{R}[\pi] = 0.5R_1[\pi] + 0.5R_2[\pi] - \beta\mathrm{KL}[\pi \mid \pi_{\mathrm{ref}}]$ with $\beta = 10^{-4}$ using the non-linear GRPO algorithm. Here, $R_i[\pi] = \mathbb{E}_{y \sim \mathcal{T}_i[\pi]}[r_i(y)]$ with $\mathcal{T}_i$ being BoN or soft BoN sampling. As a baseline, we also train the base model using standard GRPO to maximize the average expected reward $\mathbb{E}_\pi[(r_1(y) + r_2(y))/2]$ - $\beta\mathrm{KL}[\pi \mid \pi_{\mathrm{ref}}]$.

Fig. 3a (left) shows the response length distributions generated by the IAMA models for BoN objective ($N = 2, 4$) and Fig. 3a (right) shows those for soft BoN sampling ($\tau = 0.09, 0.1$). Note that this is the base model distribution, i.e., the distribution without (soft) BoN sampling. We can see that the meta-aligned model produces diverse responses that cover both short and long lengths effectively while the baseline model generates only medium-length responses. In particular, larger $N$ in BoN and smaller $\tau$ in soft BoN lead to stronger polarization of response lengths. We also experimented with BoN with $N = 8, 16$, which are equal or greater than the number of sampled responses during training ($M = 8$) and confirmed that non-linear GRPO works well even in such cases. The results are provided in Appendix J.2 due to space limitations.

### 6.2. RLHF with helpfulness and harmlessness

Next, we evaluate the proposed method on the RLHF task using hh-rlhf dataset (Bai et al., 2022). We optimize a reproduced version of Alpaca 7B (Taori et al., 2023; Dai et al., 2024) model to maximize $w_1 R_{\mathrm{helpful}}[\pi] + (1 - w_1)R_{\mathrm{harmless}}[\pi]$, where $R_{\mathrm{helpful}}[\pi]$ and $R_{\mathrm{harmless}}[\pi]$ are BoN objectives with $N = 4$ for helpfulness and harmlessness rewards, respectively. We vary the weight $w_1$ to see how well the model can trade off between helpfulness and harmlessness. As reward models, we train Qwen3 4B (Yang et al., 2025) based on the Bradley-Terry objective (Bradley & Terry, 1952). Since evaluation with human raters can be

costly, we use the larger reward models based on Qwen 32B as golden rewards to evaluate the aligned models following previous works (Balashankar et al., 2025; Eisenstein et al., 2024).

To fairly compare the performance of different methods and weights, we fix the target KL divergence instead of fixing the regularization coefficient $\beta$. Then, we adopt log-space proportional controller (Ziegler et al., 2019) to adjust the KL-regularization coefficient $\beta$ during training. In this experiment, we set the target KL divergence as $0.1$.

Fig. 3b shows the evaluation results on helpfulness and harmlessness rewards with and without BoN sampling at inference time. Without BoN sampling, IAMA models cannot outperform standard aligned models since the standard alignment approach can directly optimize the weighted sum of expected rewards. On the other hand, with BoN sampling, IAMA models push the Pareto front forward compared to the standard alignment approach. This is because the meta-aligned model produces diverse responses covering different trade-offs between helpfulness and harmlessness, which allows the inference-time alignment to select high-reward responses effectively. We also provide the results for the case of $N = 8$ in Appendix J.3.

## 7. Conclusion

In this work, we proposed inference-aware meta-alignment (IAMA), a novel framework for aligning LLMs to diverse user preferences via two-stage alignment procedures. We formulated IAMA as a non-linear optimization problem in the space of measures and developed the non-linear GRPO algorithm to solve the problem. We proved the convexity and smoothness of the BoN-type objective and provided convergence guarantees of the proposed algorithm. While implementation is simple, experimental results demonstrated the effectiveness of IAMA via non-linear GRPO in aligning LLMs to multiple reward functions simultaneously.

## Acknowledgements

TS and RH were partially supported by JSPS KAKENHI (24K02905) and JST CREST (JPMJCR2115). This research is supported by the National Research Foundation, Singapore, Infocomm Media Development Authority under its Trust Tech Funding Initiative, and the Ministry of Digital Development and Information under the AI Visiting Professorship Programme (award number AIVP-2024-004). Any opinions, findings and conclusions or recommendations expressed in this material are those of the author(s) and do not reflect the views of National Research Foundation, Singapore, Infocomm Media Development Authority, and the Ministry of Digital Development and Information.

## Impact Statement

This work aims to advance the methodology for aligning large language models to diverse and potentially conflicting human preferences with limited computational resources at inference time. By improving the efficiency and flexibility of inference-time alignment, the proposed approach may facilitate broader deployment of aligned language models in practical applications. At the same time, as with other alignment techniques, misuse or mis-specification of alignment criteria could lead to unintended behaviors, underscoring the importance of careful design and evaluation of alignment objectives. Overall, we do not foresee unique ethical risks beyond those commonly associated with research on large language model alignment.

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

## A. Notation

For the reader's convenience, we summarize the notation used in this paper in Table 1.

*Table 1.* Summary of Notation

| Notation | Description |
|---|---|
| $\mathcal{P}$ | Set of probability distributions over the response space |
| $\pi, \pi_{\mathrm{ref}}$ | Policy and reference policy in $\mathcal{P}$ |
| $r(y)$ | Reward function mapping response $y$ to a scalar reward |
| $\mathrm{BoN}_N[\pi]$ | Best-of-N (BoN) aligned model based on base policy $\pi$ with $N$ samples |
| $\mathrm{SoftBoN}_\tau[\pi]$ | Soft BoN aligned model based on base policy $\pi$ with temperature parameter $\tau$ |
| $\frac{\partial R}{\partial \pi}[\pi](y)$ | Functional derivative of $R$ at policy $\pi$ evaluated at response $y$ |
| $\mathrm{d}R[\pi]$ | $\frac{\partial R}{\partial \pi}[\pi]$ as a linear functional over $\mathcal{P}$ |
| $\langle \mathrm{d}R[\pi], \pi' \rangle$ | Action of linear functional $\mathrm{d}R[\pi]$ on $\pi'$ defined as $\int \frac{\partial R}{\partial \pi}[\pi](y) \cdot \pi'(y)\,\mathrm{d}y$ |
| $\mathrm{KL}[\pi \mid \pi_{\mathrm{ref}}]$ | Kullback–Leibler divergence from $\pi_{\mathrm{ref}}$ to $\pi$ |
| $D_\phi[\pi \mid \mu]$ | Bregman divergence induced by convex functional $\phi$ between $\mu$ and $\pi$ |
| $\pi_r$ | Distribution of reward $r(y)$ when $y \sim \pi$ |

## B. Detailed Discussion on Related Work

In this section, we provide a more detailed discussion on related work about inference-time alignment and contrast our proposed method with prior studies. Recently, Balashankar et al. (2025) have proposed InfAlign-CTRL, which apply non-linear transformation to the reward function. However, they focus on 1) single reward function, 2) BoN-type alignment algorithms, and 3) win rate maximization objective. These assumptions are essential for deriving the method because the nonlinear transformation depends on the distribution of rewards, which can be determined from the fact that, in the case of a single reward and win-rate maximization, the distribution is uniform. Note that when there are multiple rewards, even if we consider win-rate maximization, the reward distribution is no longer uniform due to the presence of correlations.

Several works (Walder & Karkhanis, 2025; Bagirov et al., 2025) have developed unbiased gradient estimator for Pass@k (BoN) objectives to align LLMs via policy gradient methods. However, these methods cannot handle inference-time alignment algorithms beyond BoN and require $M \geq N$ samples in each gradient estimation step. Therefore, they are not applicable to large $N$ settings or Best of Poisson (BoP), which require (potentially) unbounded number of samples.

On the other hand, our proposed non-linear GRPO algorithm can handle a wide range of inference-time alignment algorithms and multiple reward functions. Moreover, even when restricted to the BoN setting, these prior studies did not investigate the theoretical properties of the optimization problem. We analyze the loss landscape of the optimization problem and establish its convexity. Based on this analysis, we provide convergence guarantees. This constitutes an additional, independent contribution.

## C. Other Non-linear Optimization Problems Related to LLM Alignment

Since our proposed non-linear GRPO algorithm is quite versatile, it can be applied to other non-linear optimization problems related to LLM alignment. In addition, non-linear GRPO can be regarded as a generalization of existing algorithms for specific non-linear optimization problems. Note that the existing works have not explicitly discussed the connection to non-linear optimization in the space of probability measures, which enables us to formulate a general algorithm and its theoretical analysis.

**Risk-aware Optimization** Conditional Value at Risk (CVaR) is a widely used risk measure in finance and reinforcement learning, which is defined as

$$\mathrm{CVaR}_\alpha[\pi] := \mathbb{E}_{y \sim \pi}\left[r(y) \mid r(y) \leq q_\alpha[\pi]\right],$$

where $q_\alpha[\pi]$ is the $\alpha$-quantile of the reward distribution when $y \sim \pi$. This objective is non-linear in $\pi$ due to the presence of the quantile function $q_\alpha[\pi]$. CPPO proposed by Ying et al. (2022) is a policy optimization algorithm for CVaR objectives,

which can be seen as a special case of our non-linear GRPO algorithm. Jiang et al. (2025) also proposed a risk-sensitive RL algorithm for LLMs, using Entropic Risk Measure (ERM) defined as

$$R_{\text{ERM}}[\pi] := \frac{1}{\beta} \log \mathbb{E}_{y \sim \pi} \left[ e^{\beta r(y)} \right].$$

If $\beta > 0$, this objective encourages risk-seeking behavior, and is concave in $\pi$ since $\log$ is concave and $\mathbb{E}_{y \sim \pi} \left[ e^{\beta r(y)} \right]$ is linear in $\pi$.

**Diversity-promoting Optimization**    Diversity-promoting optimization aims to learn a policy that generates diverse outputs. For example, Chen et al. (2025) proposed to use determinantal point processes (DPPs) as a diversity-promoting regulalrizer:

$$R_{\text{DPP}}[\pi] := \mathbb{E}_{y_1, \ldots, y_N \sim \pi} \left[ \log \det(K(y_1, \ldots, y_N)) \right],$$

where $[K(y_1, \ldots, y_N)]_{i,j} = k(y_i, y_j)$ is a kernel matrix measuring the similarity between outputs $y_1, \ldots, y_N$. Other possible regularizers include Rao's quadratic entropy:

$$R_{\text{Rao}}[\pi] := -\mathbb{E}_{y, y' \sim \pi} \left[ k(y, y') \right].$$

If kernel $k$ is positive semi-definite, the above objective is concave in $\pi$ since this can be regarded as a quadratic form induced by $k$.

# D. Practical Implementation of Non-linear GRPO

Here, we describe a practical implementation of the non-linear GRPO algorithm. This is almost identical to the standard GRPO algorithm proposed in Shao et al. (2024) except for the use of the approximated functional derivative $\frac{\partial R}{\partial \pi}[\hat{\pi}_t](y)$ as the reward function. There exist various techniques to improve the stability and efficiency of GRPO that are not listed here. They can be directly applied to our non-linear GRPO as well. Please refer to the TRL (von Werra et al., 2020) document for details.

---

**Algorithm 2** Practical Non-linear GRPO

---

1: **Input:** Initial policy $\pi_0$, reference policy $\pi_{\text{ref}}$, reward functions $\{r_i\}_{i=1}^m$, weights $\{w_i\}_{i=1}^m$, regularization parameter $\beta$, step size $\eta$, number of iterations $T$
2: **for** $t = 0, 1, \ldots, T-1$ **do**
3:     Sample $M$ responses $\{y_j\}_{j=1}^M$ from $\pi_t$
4:     Compute approximated functional derivative: $\tilde{r}_t(y) = \frac{\partial R}{\partial \pi}[\hat{\pi}_t](y)$.
5:     Compute advantages: $A_j = \frac{\tilde{r}_t(y_j) - b_t}{s_t}$ where $b_t = \frac{1}{M} \sum_{j=1}^M \tilde{r}_t(y_j)$ and $s_t = \sqrt{\frac{1}{M} \sum_{j=1}^M (\tilde{r}_t(y_j) - b_t)^2}$
6:     Define surrogate loss: $\hat{\mathcal{L}}[\pi] = \frac{1}{M} \sum_{i=1}^M \min \left[ \frac{\pi(y_i)}{\pi_t(y_i)} \cdot A_i, \text{clip}_\epsilon \left( \frac{\pi(y_i)}{\pi_t(y_i)} \right) \cdot A_i \right] + \beta \widehat{\text{KL}}[\pi \mid \pi_{\text{ref}}]$, where $\widehat{\text{KL}}[\pi \mid \pi_{\text{ref}}] = \frac{1}{M} \sum_{i=1}^M \frac{\pi_{\text{ref}}(y_i)}{\pi(y_i)} - \log \frac{\pi_{\text{ref}}(y_i)}{\pi(y_i)} - 1$
7:     Approximately solve the minimization problem $\min_\pi \hat{\mathcal{L}}[\pi]$ with optimizer (e.g., Adam) to obtain $\pi_{t+1}$
8: **end for**
9: **Return:** Final policy $\pi_T$

---

### D.1. Computation of Functional Derivative

In this section, we describe how to compute the functional derivatives for BoN and Soft BoN objectives with empirical distributions. For both cases, computational complexity depends on the number of samples $M$ instead of the size $N$ of BoN.

**BoN**    If we approximate the functional derivative $\frac{\partial R}{\partial \pi}[\hat{\pi}_t](y)$ with empirical distribution $\hat{\pi}_t$ constructed from $M$ samples $\{y_i\}_{i=1}^M$ drawn from $\pi_t$, the computational bottleneck is sorting the rewards to compute the empirical quantiles. Thus, we can compute the functional derivative at these samples in $O(M \log M)$ time as follows:

---
**Algorithm 3** Linearized Reward Computation for BoN Objective
---

1: **Input:** Reward values $\{r_j\}_{j=1}^M$, BoN parameter $N$
2: Sort indices $\{1, \ldots, M\}$ in ascending order of rewards so that $r_{(1)} \leq r_{(2)} \leq \cdots \leq r_{(M)}$, where $(i)$ denotes the index of the $i$-th smallest reward.
3: Initialize cumulative sum $c \leftarrow 0$
4: Set reference reward $r_{\text{next}} \leftarrow r_{(M)}$
5: Initialize $\tilde{r}_j \leftarrow 0$ for all $j = 1, \ldots, M$
6: **for** $i = M, M-1, \ldots, 1$ **do**
7:     Compute reward gap $\Delta r \leftarrow r_{\text{next}} - r_{(i)}$
8:     Compute empirical CDF $q \leftarrow i/M$
9:     Update cumulative contribution $c \leftarrow c + Nq^{N-1}\Delta r$
10:     Assign linearized reward $\tilde{r}_{(i)} \leftarrow -c$
11:     Update reference reward $r_{\text{next}} \leftarrow r_{(i)}$
12: **end for**
13: **Return:** $\{\tilde{r}_j\}_{j=1}^M$

---

**Soft BoN** If we approximate the functional derivative $\frac{\partial R}{\partial \pi}[\hat{\pi}_t](y)$ with empirical distribution $\hat{\pi}_t$ constructed from $M$ samples $\{y_i\}_{i=1}^M$ drawn from $\pi_t$, we can compute the functional derivative in $O(M)$ time as follows:

---
**Algorithm 4** Linearized Reward Computation for Soft BoN Objective
---

1: **Input:** Reward values $\{r_j\}_{j=1}^M$, temperature parameter $\tau$
2: Compute normalization constant $Z \leftarrow \frac{1}{M}\sum_{j=1}^M \exp(r_j/\tau)$
3: Compute softmax-weighted expected reward $\bar{r} \leftarrow \frac{1}{M}\sum_{j=1}^M r_j \exp(r_j/\tau)/Z$
4: **for** $j = 1, \ldots, M$ **do**
5:     Compute linearized reward $\tilde{r}_j \leftarrow r_j \exp(r_j/\tau)/Z - \exp(r_j/\tau)\,\bar{r}/Z$
6: **end for**
7: **Return:** Linearized rewards $\{\tilde{r}_j\}_{j=1}^M$

---

# E. Auxiliary Lemmas

**Lemma E.1.** *Let $C[\pi](r)$ be the cumulative distribution function (CDF) defined as $C[\pi](r) = \int \mathbb{1}[r(y) \leq r]\pi(y)\,\mathrm{d}y$. Then, the functional derivative of $C[\pi](r)$ is given by $\frac{\partial C}{\partial \pi}[\pi](r) = \mathbb{1}[r(y) \leq r]$.*

*Proof.* For any $\pi, \pi' \in \mathcal{P}$ and $\epsilon \in \mathbb{R}$, we have

$$C[\pi + \epsilon(\pi' - \pi)](r) - C[\pi](r) = \int \mathbb{1}[r(y) \leq r] \cdot (\pi + \epsilon(\pi' - \pi))(y)\,\mathrm{d}y - \int \mathbb{1}[r(y) \leq r] \cdot \pi(y)\,\mathrm{d}y$$

$$= \epsilon \int \mathbb{1}[r(y) \leq r] \cdot (\pi' - \pi)(y)\,\mathrm{d}y$$

and thus

$$\left.\frac{\mathrm{d}C[\pi + \epsilon(\pi' - \pi)](r)}{\mathrm{d}\epsilon}\right|_{\epsilon=0} = \int \mathbb{1}[r(y) \leq r] \cdot (\pi' - \pi)(y)\,\mathrm{d}y.$$

Hence, we have $\frac{\partial C}{\partial \pi}[\pi](r) = \mathbb{1}[r(y) \leq r]$. $\qquad\square$

**Lemma E.2.** *For a reward function $r(y)$ and a base policy $\pi$, assume that the distribution of $r(y)$ when $y \sim \pi$ is continuous. Then, the BoN policy $\text{BoN}_N[\pi]$ satisfies*

$$\text{BoN}_N[\pi](y) = N \cdot C[\pi](r(y))^{N-1} \cdot \pi(y)$$

*Proof.* See Lemma 5 of Balashankar et al. (2025). $\square$

**Lemma E.3** (Three-point inequality). *Let $\mathcal{G}, \psi, \phi$ be convex functionals. Then, for any $\mu, \nu$ and $\bar{\nu} := \arg\min \mathcal{G}[\nu] + \beta D_\psi(\nu \mid \pi_{\text{ref}}) + LD_\phi(\nu \mid \mu) \in \mathcal{P}$, we have*

$$\mathcal{G}[\bar{\nu}] + \beta D_\psi(\bar{\nu} \mid \pi_{\text{ref}}) + LD_\phi(\bar{\nu} \mid \mu) \leq \mathcal{G}[\nu] + \beta D_\psi(\nu \mid \pi_{\text{ref}}) + LD_\phi(\nu \mid \mu)$$
$$- LD_\phi(\nu \mid \bar{\nu}) - \beta D_\psi(\nu \mid \bar{\nu}).$$

*Proof.* This is an extension of Lemma 3 in Aubin-Frankowski et al. (2022) and we follow the same proof strategy. Let $f = \mathcal{G}[\nu] + \beta D_\psi(\nu \mid \pi_{\text{ref}}) + LD_\phi(\nu \mid \mu)$. Note that $f$ is convex since $\mathcal{G}$, $D_\psi$, and $D_\phi$ are convex. Then, we have

$$D_f(\nu \mid \bar{\nu}) = D_{\mathcal{G}}(\nu \mid \bar{\nu}) + \beta D_\psi(\nu \mid \bar{\nu}) + LD_\phi(\nu \mid \bar{\nu})$$
$$\geq \beta D_\psi(\nu \mid \bar{\nu}) + LD_\phi(\nu \mid \bar{\nu})$$

from the non-negativity of Bregman divergence.

By the optimality of $\bar{\nu}$, we have $D_f(\nu \mid \bar{\nu}) = f(\nu) - f(\bar{\nu}) - \langle \frac{\partial f}{\partial \pi}[\bar{\nu}], \nu - \bar{\nu} \rangle \leq f(\nu) - f(\bar{\nu})$. Thus, we have

$$f(\nu) - f(\bar{\nu}) \geq \beta D_\psi(\nu \mid \bar{\nu}) + LD_\phi(\nu \mid \bar{\nu}).$$

Rearranging the above inequality, we obtain the desired result. $\square$

## F. Proof for Propositions 4.2 and 4.3

**Case 1: BoN**  BoN defines the aligned model as

$$\text{BoN}_N[\pi](y) = f(C[\pi](r(y))) \cdot \pi(y),$$

where $f(r) = N \cdot r^{N-1}$. Thus, the objective function is given by

$$\begin{aligned}
R(\pi) &= \mathbb{E}_{y \sim \text{BoN}_N[\pi]}[r(y)] \\
&= \int r(y) \cdot f(C[\pi](r(y))) \cdot \pi(y)\, dy \\
&= \int_0^{r_{\max}} r \cdot f(C[\pi_r](r)) \cdot \pi_r(r)\, dr \\
&= [r \cdot F(C[\pi_r](r))]_0^{r_{\max}} - \int_0^{r_{\max}} F(C[\pi_r](r))\, dr
\end{aligned}$$

where $\pi_r$ is the distribution of $r(y)$ when $y \sim \pi$ and $F$ is an antiderivative of $f$. Here, the last inequality follows from integration by parts. Then, the functional derivative can be computed as

$$\begin{aligned}
\frac{\delta R[\pi]}{\delta \pi}(z) &= -\int f(C[\pi_r](r)) \frac{\delta C[\pi_r]}{\delta \pi}(z)\, dr \\
&= -\int f(C[\pi_r](r)) \cdot \mathbb{1}_{r \geq r(z)}\, dr \\
&= -\int_{r(z)}^{r_{\max}} f(C[\pi_r](r))\, dr.
\end{aligned}$$

This completes the proof.

**Case 2: Soft BoN**  Let $\lambda = 1/\tau$. Soft BoN defines the aligned model as

$$\text{SoftBoN}_\tau[\pi](y) = \frac{\exp(\lambda r(y)) \cdot \pi(y)}{\int \exp(\lambda r(z)) \cdot \pi(z)\, dz}.$$

Thus, the objective function is given by

$$R(\pi) = \mathbb{E}_{y \sim \text{SoftBoN}_\lambda} [r(y)]$$

$$= \int r(y) \cdot \frac{\exp(\lambda r(y)) \cdot \pi(y)}{\int \exp(\lambda r(z)) \cdot \pi(z) \, dz} \, dy$$

$$= \frac{d}{d\lambda} \log \left( \int \exp(\lambda r(z)) \cdot \pi(z) \, dz \right).$$

Then, the functional derivative can be computed as

$$\frac{\partial R}{\partial \pi}[\pi](y) = \frac{d}{d\lambda} \left( \frac{\partial}{\partial \pi} \log \left( \int \exp(\lambda r(z)) \cdot \pi(z) \, dz \right) \right)$$

$$= \frac{d}{d\lambda} \left( \frac{\exp(\lambda r(y))}{\int \exp(\lambda r(z)) \cdot \pi(z) \, dz} \right)$$

$$= \frac{r(y) \exp(\lambda r(y))}{\int \exp(\lambda r(z)) \cdot \pi(z) \, dz} - \frac{\exp(\lambda r(y)) \cdot \int r(z) \exp(\lambda r(z)) \cdot \pi(z) \, dz}{\left( \int \exp(\lambda r(z)) \cdot \pi(z) \, dz \right)^2}.$$

This completes the proof.

## G. Proof for Section 3

### G.1. Proof of Proposition 3.1

Let $Y_1, \ldots, Y_N \overset{\text{i.i.d.}}{\sim} \pi$ be $N$ i.i.d. samples from $\pi$. Since $r_1$ is strictly decreasing and $r_2$ is strictly increasing, the samples maximizing $r_1$ and $r_2$ are the minimum and maximum order statistics,

$$Y_{(1)} := \min_{k \leq N} Y_k,$$

$$Y_{(N)} := \max_{k \leq N} Y_k.$$

Hence, the objective is

$$R[\pi] := \mathbb{E}[-Y_{(1)}^2] + \mathbb{E}[-(1 - Y_{(N)})^2].$$

For any $y \in [0, 1]$,

$$\text{Prob} \left( Y_{(1)} > y \right) = (1 - C[\pi](y))^N,$$

$$\text{Prob} \left( Y_{(N)} \leq y \right) = C[\pi](y)^N.$$

Using $\mathbb{E}[X^2] = \int_0^1 2t \, \mathbb{P}(X > t) \, dt$ for $X \in [0, 1]$ and a change of variables,

$$R[\pi] = -2 \int_0^1 \left[ y(1 - C[\pi](y))^N + (1 - y)C[\pi](y)^N \right] dy.$$

Thus maximizing $R[\pi]$ is equivalent to minimizing

$$I(\pi) := \int_0^1 \phi_y(C[\pi](y)) \, dy,$$

where $\phi_y(p) := y(1 - p)^N + (1 - y)p^N$. For fixed $y \in (0, 1)$, we have

$$\phi_y''(p) = N(N - 1)\left[ y(1 - p)^{N-2} + (1 - y)p^{N-2} \right] > 0$$

for $p \in (0, 1)$. Therefore, $\phi_y$ is strictly convex on $(0, 1)$ and has a unique minimizer satisfying

$$0 = \phi'_y(p) = -Ny(1 - p)^{N-1} + N(1 - y)p^{N-1}$$
$$\iff \left(\frac{p}{1 - p}\right)^{N-1} = \frac{y}{1 - y}.$$

That is, the minimizer is given by

$$C^*(y) = \frac{y^\alpha}{y^\alpha + (1 - y)^\alpha}, \qquad \alpha := \frac{1}{N - 1}.$$

Moreover, $C^*(y)$ is strictly increasing with $C^*(0) = 0$ and $C^*(1) = 1$, which implies that $\pi_* := \frac{dC^*}{dy}$ is in $\mathcal{P}$ and a minimizer of $I(\pi)$. This completes the proof.

## H. Proofs for Section 5

### H.1. Proof of Theorem 5.2

For notational simplicity, we define

$$F[\pi] := -R[\pi],$$
$$\mathcal{L}[\pi] = F[\pi] + \beta\mathrm{KL}[\pi \mid \pi_{\mathrm{ref}}].$$

Note that $F$ is convex and $L$-smooth since $R$ is concave and $L$-smooth.

From the $L$-smoothness of $F[\pi]$, we have

$$\mathcal{L}[\pi_{t+1}] - \mathcal{L}[\pi_t] \leq \langle \mathrm{d}F[\pi_t], \pi_{t+1} - \pi_t \rangle + L\mathrm{KL}[\pi_{t+1} \mid \pi_t] + \beta\mathrm{KL}[\pi_{t+1} \mid \pi_{\mathrm{ref}}] - \beta\mathrm{KL}[\pi_t \mid \pi_{\mathrm{ref}}].$$

Applying Lemma E.3 to the convex functional $\langle \mathrm{d}F[\pi_t], \pi - \pi_t \rangle$, we obtain

$$\langle \mathrm{d}F[\pi_t], \pi_{t+1} - \pi_t \rangle + L\mathrm{KL}[\pi_{t+1} \mid \pi_t] + \beta\mathrm{KL}[\pi_{t+1} \mid \pi_{\mathrm{ref}}] \leq \langle \mathrm{d}F[\pi_t], \pi - \pi_t \rangle + L\mathrm{KL}[\pi \mid \pi_t] + \beta\mathrm{KL}[\pi \mid \pi_{\mathrm{ref}}]$$
$$- (L + \beta)\mathrm{KL}[\pi \mid \pi_{t+1}]$$

for any $\pi \in \mathcal{P}$. Combining the above two inequalities, the following inequality holds:

$$\mathcal{L}[\pi_{t+1}] - \mathcal{L}[\pi_t] \leq \langle \mathrm{d}F[\pi_t], \pi - \pi_t \rangle$$
$$+ L\mathrm{KL}[\pi \mid \pi_t]$$
$$+ \beta\mathrm{KL}[\pi \mid \pi_{\mathrm{ref}}]$$
$$- (L + \beta)\mathrm{KL}[\pi \mid \pi_{t+1}]$$
$$- \beta\mathrm{KL}[\pi_t \mid \pi_{\mathrm{ref}}].$$

By setting $\pi = \pi_t$, we have

$$\mathcal{L}[\pi_{t+1}] - \mathcal{L}[\pi_t] \leq -(L + \beta)\mathrm{KL}[\pi_t \mid \pi_{t+1}] \leq 0,$$

which implies the monotonicity of $\mathcal{L}[\pi_t]$.

On the other hand, by setting $\pi = \pi_*$, we have

$$\mathcal{L}[\pi_{t+1}] - \mathcal{L}[\pi_t] \leq \langle \mathrm{d}F[\pi_t], \pi_* - \pi_t \rangle$$
$$+ L\mathrm{KL}[\pi_* \mid \pi_t]$$
$$+ \beta\mathrm{KL}[\pi_* \mid \pi_{\mathrm{ref}}]$$
$$- (L + \beta)\mathrm{KL}[\pi_* \mid \pi_{t+1}]$$
$$- \beta\mathrm{KL}[\pi_t \mid \pi_{\mathrm{ref}}].$$

Since $F$ is convex, we have

$$\langle \mathrm{d}F[\pi_t], \pi_* - \pi_t \rangle \leq F(\pi_*) - F(\pi_t).$$

Thus, we obtain

$$\begin{aligned}
\mathcal{L}[\pi_{t+1}] - \mathcal{L}[\pi_t] &\leq F(\pi_*) - F(\pi_t) \\
&\quad + L\mathrm{KL}[\pi_* \mid \pi_t] \\
&\quad + \beta\mathrm{KL}[\pi_* \mid \pi_{\mathrm{ref}}] \\
&\quad - (L + \beta)\mathrm{KL}[\pi_* \mid \pi_{t+1}] \\
&\quad - \beta\mathrm{KL}[\pi_t \mid \pi_{\mathrm{ref}}] \\
&= \mathcal{L}[\pi_*] - \mathcal{L}[\pi_t] \\
&\quad + L\mathrm{KL}[\pi_* \mid \pi_t] \\
&\quad - (L + \beta)\mathrm{KL}[\pi_* \mid \pi_{t+1}].
\end{aligned}$$

Rearranging the above inequality, we have

$$\mathcal{L}[\pi_{t+1}] \leq \mathcal{L}[\pi_*] + L\mathrm{KL}[\pi_* \mid \pi_t] - (L + \beta)\mathrm{KL}[\pi_* \mid \pi_{t+1}].$$

Summing the above inequality from $t = 0$ to $T - 1$, we obtain

$$\begin{aligned}
\sum_{t=1}^{T} \left(\frac{L+\beta}{L}\right)^t \mathcal{L}[\pi_t] &\leq \sum_{t=1}^{T} \left(\frac{L+\beta}{L}\right)^t \mathcal{L}[\pi_*] + L\sum_{t=1}^{T} \left(\frac{L+\beta}{L}\right)^t \mathrm{KL}[\pi_* \mid \pi_{t-1}] - (L+\beta)\sum_{t=1}^{T} \left(\frac{L+\beta}{L}\right)^t \mathrm{KL}[\pi_* \mid \pi_t] \\
&= \sum_{t=1}^{T} \left(\frac{L+\beta}{L}\right)^t \mathcal{L}[\pi_*] + (L+\beta)\left[\sum_{t=1}^{T} \left(\frac{L+\beta}{L}\right)^{t-1} \mathrm{KL}[\pi_* \mid \pi_{t-1}] - \sum_{t=1}^{T} \left(\frac{L+\beta}{L}\right)^t \mathrm{KL}[\pi_* \mid \pi_t]\right] \\
&= \sum_{t=1}^{T} \left(\frac{L+\beta}{L}\right)^t \mathcal{L}[\pi_*] + (L+\beta)\left[\sum_{t=0}^{T-1} \left(\frac{L+\beta}{L}\right)^t \mathrm{KL}[\pi_* \mid \pi_t] - \sum_{t=1}^{T} \left(\frac{L+\beta}{L}\right)^t \mathrm{KL}[\pi_* \mid \pi_t]\right] \\
&= \sum_{t=1}^{T} \left(\frac{L+\beta}{L}\right)^t \mathcal{L}[\pi_*] + (L+\beta)\left[\mathrm{KL}[\pi_* \mid \pi_0] - \left(\frac{L+\beta}{L}\right)^T \mathrm{KL}[\pi_* \mid \pi_T]\right] \\
&\leq \sum_{t=1}^{T} \left(\frac{L+\beta}{L}\right)^t \mathcal{L}[\pi_*] + (L+\beta)\mathrm{KL}[\pi_* \mid \pi_0].
\end{aligned}$$

Dividing both sides by $\sum_{t=1}^{T} \left(\frac{L+\beta}{L}\right)^t$, we have

$$\begin{aligned}
\mathcal{L}[\pi_T] - \mathcal{L}[\pi_*] &\leq \frac{\sum_{t=1}^{T} \left(\frac{L+\beta}{L}\right)^t \mathcal{L}[\pi_t]}{\sum_{t=1}^{T} \left(\frac{L+\beta}{L}\right)^t} - \mathcal{L}[\pi_*] \\
&\leq \frac{(L+\beta)\mathrm{KL}[\pi_* \mid \pi_0]}{\sum_{t=1}^{T} \left(\frac{L+\beta}{L}\right)^t} \\
&= \frac{(L+\beta)\mathrm{KL}[\pi_* \mid \pi_0]}{\frac{(L+\beta)}{L} \cdot \frac{((L+\beta)/L)^T - 1}{(L+\beta)/L - 1}} \\
&= \frac{\beta \cdot \mathrm{KL}[\pi_* \mid \pi_0]}{((L+\beta)/L)^T - 1}.
\end{aligned}$$

The first inequality follows from the monotonicity of $\mathcal{L}[\pi_t]$. This completes the proof.

## H.2. Proof of Theorem 5.3

Let the empirical gradient be $\hat{g}_t := \mathrm{d}F[\hat{\pi}_t]$ and define the estimation error $e_t := \hat{g}_t - \mathrm{d}F[\pi_t]$. From the $L$-smoothness of $F$, we have

$$\mathcal{L}[\pi_{t+1}] - \mathcal{L}[\pi_t] \leq \mathrm{d}F[\pi_t](\pi_{t+1} - \pi_t) + L\,\mathrm{KL}[\pi_{t+1} \mid \pi_t] + \beta\,\mathrm{KL}[\pi_{t+1} \mid \pi_{\mathrm{ref}}] - \beta\,\mathrm{KL}[\pi_t \mid \pi_{\mathrm{ref}}]$$
$$= \hat{g}_t(\pi_{t+1} - \pi_t) + L\,\mathrm{KL}[\pi_{t+1} \mid \pi_t] + \beta\,\mathrm{KL}[\pi_{t+1} \mid \pi_{\mathrm{ref}}] - \beta\,\mathrm{KL}[\pi_t \mid \pi_{\mathrm{ref}}] - e_t(\pi_{t+1} - \pi_t).$$

Due to the inexact optimization, there exists a residual term $r_t$ defined as

$$r_t := \hat{g}_t + \frac{1}{\eta}\log\frac{\pi_{t+1}}{\pi_t} + \beta\log\frac{\pi_{t+1}}{\pi_{\mathrm{ref}}}.$$

Note that $r_t$ is constant if the optimization is exact. This implies that $\pi_{t+1}$ is the optimal solution to the following optimization problem:

$$\min_{\pi \in \mathcal{P}}\langle\hat{g}_t - r_t, \pi - \pi_t\rangle + \frac{1}{\eta}\mathrm{KL}[\pi \mid \pi_t] + \beta\mathrm{KL}[\pi \mid \pi_{\mathrm{ref}}].$$

Applying Lemma E.3 to the convex functional $\langle\hat{g}_t - r_t, \pi - \pi_t\rangle$, we obtain

$$\langle\hat{g}_t - r_t, \pi_{t+1} - \pi_t\rangle + L\mathrm{KL}[\pi_{t+1} \mid \pi_t] + \beta\mathrm{KL}[\pi_{t+1} \mid \pi_{\mathrm{ref}}]$$
$$= \langle\hat{g}_t, \pi_{t+1} - \pi_t\rangle + L\mathrm{KL}[\pi_{t+1} \mid \pi_t] + \beta\mathrm{KL}[\pi_{t+1} \mid \pi_{\mathrm{ref}}] - \langle r_t, \pi_{t+1} - \pi_t\rangle$$
$$\leq \langle\hat{g}_t - r_t, \pi_* - \pi_t\rangle + L\mathrm{KL}[\pi_* \mid \pi_t] + \beta\mathrm{KL}[\pi_* \mid \pi_{\mathrm{ref}}] - (L + \beta)\mathrm{KL}[\pi_* \mid \pi_{t+1}]$$
$$= \langle\hat{g}_t, \pi_* - \pi_t\rangle + L\mathrm{KL}[\pi_* \mid \pi_t] + \beta\mathrm{KL}[\pi_* \mid \pi_{\mathrm{ref}}] - (L + \beta)\mathrm{KL}[\pi_* \mid \pi_{t+1}] - \langle r_t, \pi_* - \pi_t\rangle.$$

Combining the above two inequalities, we have

$$\mathcal{L}[\pi_{t+1}] - \mathcal{L}[\pi_t] \leq \hat{g}_t(\pi_* - \pi_t) + L\mathrm{KL}[\pi_* \mid \pi_t] + \beta\mathrm{KL}[\pi_* \mid \pi_{\mathrm{ref}}]$$
$$- (L + \beta)\mathrm{KL}[\pi_* \mid \pi_{t+1}] - \beta\mathrm{KL}[\pi_t \mid \pi_{\mathrm{ref}}] - \langle e_t, \pi_{t+1} - \pi_t\rangle - \langle r_t, \pi_* - \pi_{t+1}\rangle. \tag{2}$$

By convexity of $F$,

$$\mathrm{d}F[\pi_t](\pi_* - \pi_t) \leq F[\pi_*] - F[\pi_t],$$

and thus

$$\hat{g}_t(\pi_* - \pi_t) = \mathrm{d}F[\pi_t](\pi_* - \pi_t) + \langle e_t, \pi_* - \pi_t\rangle \leq F[\pi_*] - F[\pi_t] + \langle e_t, \pi_* - \pi_t\rangle.$$

Substituting this into (2), we get

$$\mathcal{L}[\pi_{t+1}] - \mathcal{L}[\pi_t] \leq F[\pi_*] - F[\pi_t] + L\mathrm{KL}[\pi_* \mid \pi_t] + \beta\mathrm{KL}[\pi_* \mid \pi_{\mathrm{ref}}]$$
$$- (L + \beta)\mathrm{KL}[\pi_* \mid \pi_{t+1}] - \beta\mathrm{KL}[\pi_t \mid \pi_{\mathrm{ref}}] + \langle e_t, \pi_* - \pi_{t+1}\rangle - \langle r_t, \pi_* - \pi_{t+1}\rangle.$$

Rearranging the above inequality, we have

$$\mathcal{L}[\pi_{t+1}] \leq \mathcal{L}[\pi_*] + L\mathrm{KL}[\pi_* \mid \pi_t] - (L + \beta)\mathrm{KL}[\pi_* \mid \pi_{t+1}] + \langle e_t, \pi_* - \pi_{t+1}\rangle - \langle r_t, \pi_* - \pi_{t+1}\rangle. \tag{3}$$

Let $c = (\sup_y e_t(y) + \inf_y e_t(y))/2$. By Pinsker's inequality and Hölder's inequality,

$$\langle e_t, \pi_* - \pi_{t+1}\rangle = \langle e_t - c, \pi_* - \pi_{t+1}\rangle$$
$$\leq \|e_t - c\|_\infty\|\pi_* - \pi_{t+1}\|_1$$
$$= \|e_t\|_{\mathrm{sp}} \cdot \|\pi_* - \pi_{t+1}\|_1$$
$$\leq \|e_t\|_{\mathrm{sp}} \cdot \sqrt{2\,\mathrm{KL}[\pi_* \mid \pi_{t+1}]}.$$

The second equality follows from the fact $\|e_t - c\|_\infty = \max(\sup_y e_t(y) - c, c - \inf_y e_t(y)) = (\sup_y e_t(y) - \inf_y e_t(y))/2 = \|e_t\|_{\mathrm{sp}}$. Applying Young's inequality ($ab \leq \frac{a^2}{2\alpha} + \frac{\alpha b^2}{2}$), we obtain

$$\langle e_t, \pi_* - \pi_{t+1} \rangle \leq \frac{\|e_t\|_{\mathrm{sp}}^2}{2\alpha} + \alpha \mathrm{KL}[\pi_* \mid \pi_{t+1}]$$

for any $\alpha > 0$.

Similarly, we have

$$-\langle r_t, \pi_* - \pi_{t+1} \rangle \leq \frac{\|r_t\|_{\mathrm{sp}}^2}{2\alpha} + \alpha \mathrm{KL}[\pi_* \mid \pi_{t+1}]$$

for any $\alpha > 0$.

Substituting these into (3), we obtain

$$\mathcal{L}[\pi_{t+1}] \leq \mathcal{L}[\pi_*] + L \, \mathrm{KL}[\pi_* \mid \pi_t] - (L + \beta - 2\alpha) \, \mathrm{KL}[\pi_* \mid \pi_{t+1}] + \frac{\|e_t\|_{\mathrm{sp}}^2 + \|r_t\|_{\mathrm{sp}}^2}{2\alpha}. \tag{4}$$

Setting $\alpha = \beta/4$ (so that $\beta - 2\alpha = \beta/2$), we have

$$\mathcal{L}[\pi_{t+1}] \leq \mathcal{L}[\pi_*] + L\mathrm{KL}[\pi_* \mid \pi_t] - (L + \beta/2) \, \mathrm{KL}[\pi_* \mid \pi_{t+1}] + \frac{2(\|e_t\|_{\mathrm{sp}}^2 + \|r_t\|_{\mathrm{sp}}^2)}{\beta}.$$

Let $\mu := L + \beta/2$. Summing over $t = 1, \ldots, T$ with weights $(\mu/L)^t$ yields

$$\sum_{t=1}^T \left(\frac{\mu}{L}\right)^t \mathcal{L}[\pi_t] \leq \sum_{t=1}^T \left(\frac{\mu}{L}\right)^t \mathcal{L}[\pi_*] + \mu \left[ \mathrm{KL}[\pi_* \mid \pi_0] - \left(\frac{\mu}{L}\right)^T \mathrm{KL}[\pi_* \mid \pi_T] \right] + \sum_{t=1}^T \left(\frac{\mu}{L}\right)^t \frac{2(\|e_{t-1}\|_{\mathrm{sp}}^2 + \|r_{t-1}\|_{\mathrm{sp}}^2)}{\beta}.$$

Let $S_T := \sum_{t=1}^T (\mu/L)^t = \frac{\mu}{L} \cdot \frac{(\mu/L)^T - 1}{(\mu/L) - 1}$. Then

$$\frac{\sum_{t=1}^T \left(\frac{\mu}{L}\right)^t \mathcal{L}[\pi_t]}{S_T} - \mathcal{L}[\pi_*] = \sum_{t=1}^T \mathrm{Prob}\left(\hat{t} = t\right) \mathcal{L}[\pi_t] - \mathcal{L}[\pi_*]$$

$$\leq \frac{\mu \, \mathrm{KL}[\pi_* \mid \pi_0]}{S_T} + \frac{2}{\beta} \cdot \frac{\sum_{t=1}^T (\mu/L)^t (\|e_{t-1}\|_{\mathrm{sp}}^2 + \|r_{t-1}\|_{\mathrm{sp}}^2)}{S_T}.$$

Substituting $\mu = L + \beta/2$ gives

$$\frac{\mu \, \mathrm{KL}[\pi_* \mid \pi_0]}{S_T} = \frac{(L + \beta/2) \, \mathrm{KL}[\pi_* \mid \pi_0]}{\frac{(L+\beta/2)}{L} \cdot \frac{((L+\beta/2)/L)^T - 1}{((L+\beta/2)/L) - 1}}$$

$$= \frac{(\beta/2) \cdot \mathrm{KL}[\pi_* \mid \pi_0]}{((L + \beta/2)/L)^T - 1}.$$

From the assumptions $\mathbb{E}[\|e_t\|_{\mathrm{sp}}^2] \leq \varepsilon$ and $\mathbb{E}[\|r_t\|_{\mathrm{sp}}^2] \leq \delta$, we have

$$\mathbb{E}\left[ \frac{\sum_{t=1}^T (\mu/L)^t (\|e_{t-1}\|_{\mathrm{sp}}^2 + \|r_{t-1}\|_{\mathrm{sp}}^2)}{S_T} \right] \leq \frac{\sum_{t=1}^T (\mu/L)^t (\varepsilon + \delta)}{S_T} = \varepsilon + \delta.$$

Thus, we have

$$\mathbb{E}\left[ \mathcal{L}[\pi_{\hat{t}}] - \mathcal{L}[\pi_*] \right] \leq \frac{(\beta/2) \cdot \mathrm{KL}[\pi_* \mid \pi_0]}{((L + \beta/2)/L)^T - 1} + \frac{2(\varepsilon + \delta)}{\beta}.$$

**H.3. Proof of Lemma 5.4**

*Proof.* Let $F$ be the antiderivative of $f$. Since $f$ is non-decreasing, $F$ is convex. By integration by parts, we have

$$
\begin{aligned}
R[\pi] &= \int r(y) f(C[\pi](r(y))) \pi(y) \, \mathrm{d}y \\
&= \int_0^\infty r f(C[\pi](r)) \pi_r(r) \, \mathrm{d}r \\
&= \int_0^\infty r \cdot \frac{\mathrm{d}}{\mathrm{d}r} F(C[\pi](r)) \, \mathrm{d}r \\
&= [r F[\pi](r)]_0^{r_{\max}} - \int_0^{r_{\max}} F(C[\pi](r)) \, \mathrm{d}r \\
&= r_{\max} F(1) - \int_0^{r_{\max}} F(C[\pi](r)) \, \mathrm{d}r.
\end{aligned}
$$

Thus, it is suffice to show that $P[\pi] := \int_0^{r_{\max}} F(C[\pi](r)) \, \mathrm{d}r$ is convex in $\pi$. For any $\pi, \pi' \in \mathcal{P}$ and $\lambda \in [0, 1]$, we have

$$
\begin{aligned}
P[(\lambda \pi + (1 - \lambda) \pi')] &= \int_0^{r_{\max}} F(C[(\lambda \pi + (1 - \lambda) \pi')](r)) \, \mathrm{d}r \\
&= \int_0^{r_{\max}} F\left( \lambda C[\pi](r) + (1 - \lambda) C[\pi'](r) \right) \, \mathrm{d}r \\
&\leq \int_0^{r_{\max}} \left( \lambda F(C[\pi](r)) + (1 - \lambda) F(C[\pi'](r)) \right) \, \mathrm{d}r \\
&= \lambda P[\pi] + (1 - \lambda) P[\pi'].
\end{aligned}
$$

Here, the inequality follows from the convexity of $F$. This completes the proof. $\qquad\square$

**H.4. Proof of Lemma 5.5**

*Proof.* From the integration by parts, we have

$$
\begin{aligned}
R(\pi) &= \int_0^{r_{\max}} r(y) f(C[\pi](r)) \, \mathrm{d}\pi(y) \\
&= [r F(C[\pi](r))]_0^{r_{\max}} - \int_0^{r_{\max}} F(C[\pi](r)) \, \mathrm{d}r, \\
&= r_{\max} F(1) - \int_0^{r_{\max}} F(C[\pi](r)) \, \mathrm{d}r,
\end{aligned}
$$

where $F$ is the antiderivative of $f$. Since the first term is constant, it suffices to show that $\int_0^{r_{\max}} F(C[\pi](r)) \, \mathrm{d}r$ is $L_f \cdot r_{\max}$-smooth relative to the KL-divergence.

First, we prove $F(C[\pi](r))$ is $L_f$-smooth relative to the KL-divergence for all $r$. For any $\pi, \pi'$, we have

$$F(C[\pi'](r)) - F(C[\pi](r)) - \int f(C[\pi](r)) \cdot \mathbb{1}_{r(y) \leq r} \cdot (\pi'(y) - \pi(y)) \, \mathrm{d}y$$

$$= F(C[\pi'](r)) - F(C[\pi](r)) - f(C[\pi](r)) \cdot (C[\pi'](r) - C[\pi](r))$$

$$\leq \frac{L_f}{2} (C[\pi'](r) - C[\pi](r))^2$$

$$= \frac{L_f}{2} \left( \int (\pi'(y) - \pi(y)) \cdot \mathbb{1}_{r(y) \leq r} \, \mathrm{d}y \right)^2$$

$$\leq \frac{L_f}{2} \left( \int |\pi'(y) - \pi(y)| \cdot \mathbb{1}_{r(y) \leq r} \, \mathrm{d}y \right)^2$$

$$\leq \frac{L_f}{2} \left( \int |\pi'(y) - \pi(y)| \, \mathrm{d}y \right)^2$$

$$\leq 2L_f d_{\mathrm{TV}}(\pi, \pi')^2$$

$$\leq L_f \mathrm{KL}[\pi' \mid \pi].$$

Here, the first inequality follows from the $L_f$-smoothness of $F$, and the last inequality follows from Pinsker's inequality. Thus, $F(C[\pi](r))$ is $L_f$-smooth relative to the KL-divergence for all $r$.

Next, by integrating both sides over $r \in [0, r_{\max}]$, we have

$$\int_0^{r_{\max}} F(C[\pi'](r)) \, \mathrm{d}r - \int_0^{r_{\max}} F(C[\pi](r)) \, \mathrm{d}r - \int_0^{r_{\max}} \int f(C[\pi](r)) \cdot \mathbb{1}_{r(y) \leq r} \cdot (\pi'(y) - \pi(y)) \, \mathrm{d}y \, \mathrm{d}r$$

$$\leq L_f \cdot r_{\max} \cdot \mathrm{KL}[\pi' \mid \pi].$$

Thus, $\int_0^{r_{\max}} F(C[\pi](r)) \, \mathrm{d}r$ is $L_f \cdot r_{\max}$-smooth relative to the KL-divergence. $\qquad\square$

### H.5. Proof of Lemma 5.6

Since the span seminorm satisfies $\|f\|_{\mathrm{sp}} \leq \|f\|_\infty$, it suffices to bound the infinity norm. The first-order variation can be computed as

$$\frac{\partial R}{\partial \pi}[\pi](y) = -\int_{r(y)}^{r_{\max}} f(C[\pi](r)) \, \mathrm{d}r.$$

Thus, the approximation error can be bounded as follows:

$$\left| \frac{\partial R}{\partial \pi}[\hat{\pi}_t](y) - \frac{\partial R}{\partial \pi}[\pi_t](y) \right| \leq \int_0^{r_{\max}} |f(C[\hat{\pi}_t](r)) - f(C[\pi_t](r))| \, \mathrm{d}r$$

$$\leq L_f r_{\max} \|C[\hat{\pi}_t] - C[\pi_t]\|_\infty$$

Let $Z = \|C[\hat{\pi}_t] - C[\pi_t]\|_\infty$ be a random variable. Using Dvoretzky-Kiefer-Wolfowitz (DKW) inequality, we have

$$\mathrm{Prob}\,(Z \geq \varepsilon) \leq 2e^{-2M\varepsilon^2},$$

which implies

$$\mathbb{E}\left[Z^2\right] = 2 \int_0^1 \varepsilon \mathrm{Prob}\,(Z \geq \varepsilon) \, \mathrm{d}\varepsilon \leq 2 \int_0^1 \varepsilon \cdot 2e^{-2M\varepsilon^2} \, \mathrm{d}\varepsilon$$

$$\leq \frac{1}{M}.$$

Thus, we have

$$\mathbb{E}\left[ \left\| \frac{\partial R}{\partial \pi}[\hat{\pi}_t](y) - \frac{\partial R}{\partial \pi}[\pi_t](y) \right\|_\infty^2 \right] \leq \frac{L_f^2 r_{\max}^2}{M},$$

which completes the proof.

# I. Experimental Details

Here, we provide additional details regarding our experiments. Hyperparameters used in our experiments are summarized in Table 2.

**Compute Resources**   Our experiments were conducted on Intel(R) Xeon(R) Silver 4316 CPU @ 2.30GHz and 8 NVIDIA A100-SXM4-80GB GPUs.

**Length Reward Experiments**   We set the number of responses $M = 8$, learning rate $1e - 6$, batch size $48$, number of iterations $T = 3000$, and $\beta = 1e - 4$. We use constant learning rate schedule and turn off reward scaling in GRPO to stabilize training. Other hyperparameters are default values from the TRL library (von Werra et al., 2020).

**HH-RLHF Experiments**   We set the number of responses $M = 8$, learning rate $1e - 6$, batch size $48$, number of iterations $T = 1000$, and the initial $\beta = 1e - 2$. We adopt log-space proportional controller (Ziegler et al., 2019) to adjust the KL-regularization coefficient $\beta$ during training as follows:

$$e_t = \text{clip}\left(\frac{\text{KL}[\pi_t \mid \pi_{\text{ref}}] - \text{KL}_{\text{target}}}{\text{KL}_{\text{target}}}, [-0.2, 0.2]\right),$$
$$\beta_{t+1} = \beta_t \cdot (1 + 0.1 \cdot e_t)$$

with target KL divergence $\text{KL}_{\text{target}} = 0.1$. To stabilize training, we turn off reward scaling in GRPO. Other hyperparameters are default values from the TRL library (von Werra et al., 2020). For evaluation, we use the first 1000 samples from the validation set of the HH-RLHF dataset to reduce the computational cost.

*Table 2.* Hyperparameters used in our experiments.

| Hyperparameter | Length Reward Experiments | HH-RLHF Experiments |
|---|---|---|
| Number of responses $M$ | 8 | 8 |
| Learning rate | $10^{-6}$ | $10^{-6}$ |
| Per device batch size | 8 | 8 |
| Number of iterations $T$ | 3000 | 1000 |
| Initial KL coefficient $\beta$ | $10^{-4}$ | $10^{-2}$ |
| Target KL $\text{KL}_{\text{target}}$ | N/A | $10^{-1}$ |
| Learning rate scheduler | Constant | Linear |
| bf16 Training | True | True |
| Optimizer | AdamW | AdamW |
| Clipping $\epsilon$ | 0.2 | 0.2 |

# J. Additional Experimental Results

## J.1. Effect of the Number of Responses $M$

To investigate the effect of the number of responses $M$ in the estimation of linearized loss, we conduct additional synthetic experiments. Since it is difficult to obtain true expected reward with inference-time alignment in general, we consider tractable setup where the reward distribution is a beta distribution on $[0, 1]$ and the inference-time alignment is performed by BoN with $N$ responses. Let $\pi_\alpha(y)$ be a beta distribution with shape parameters $(\alpha, 1)$ on $[0, 1]$. Then, $\pi_1(y)$ is equivalent to the uniform distribution on $[0, 1]$. As is well known, $\text{BoN}_N[\pi_\alpha](y)$ is a beta distribution $\pi_{\alpha N}(y)$ and its expected reward is given by $R_{\text{bon}}[\pi_1] = \frac{\alpha N}{\alpha N + 1}$. Therefore, we can analytically compute the true difference of expected reward as $R_{\text{bon}}[\pi_\alpha] - R_{\text{bon}}[\pi_1] = \frac{N\alpha}{N\alpha + 1} - \frac{N}{N+1}$. We expect that the estimated linearized loss $\langle \text{d}R[\hat{\pi}], \pi_\alpha - \pi_1 \rangle$ approximates this true difference well when $M$ is sufficiently large and $\alpha \simeq 1$. Specifically, we construct $\hat{\pi}$ by drawing $M = 2, \ldots, 32$ samples from $\pi_1$ and estimate the linearized loss for $\alpha = 1 + 10^{-4}$. Then, the mean squared error between the estimated linearized loss and the true difference of expected reward is computed over 100,000 trials. To reduce the variance, linearized

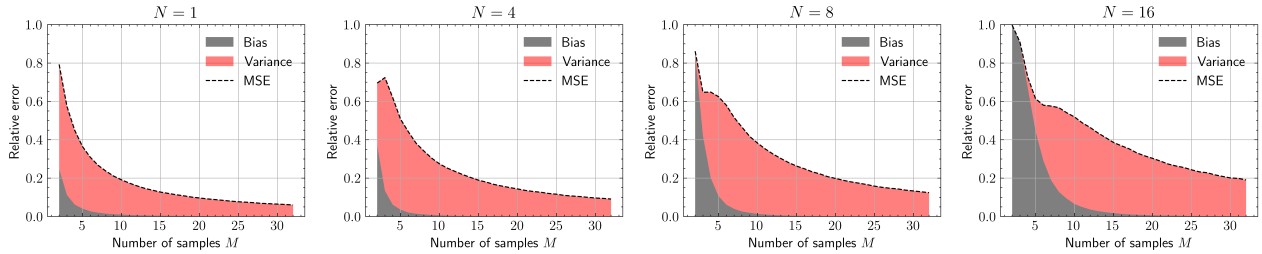

*Figure 4.* Expected error of linearized loss estimation with different $M$ and $N$ values. From left to right, $N = 1, 4, 8, 16$. The shaded area indicates the bias-variance decomposition of the expected error.

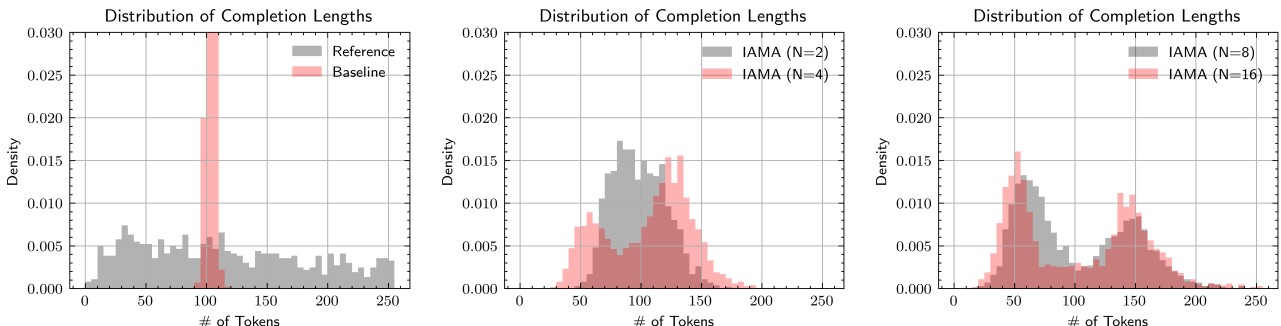

*Figure 5.* Distribution of completion lengths of reference model $\pi_{\mathrm{ref}}$, baseline model aligned by standard GRPO, and our IAMA models trained with BoN ($N = 2, 4, 8, 16$) objectives.

reward is centered by subtracting the mean as in GRPO algorithm. Figure 4 shows the comparison results for different $M$ and $N$ values. We can observe that large $N$ requires larger $M$ to achieve low expected error but it still achieves reasonable accuracy even with $M < N$ compared to $N = 1$ case, which corresponds to the standard GRPO. In particular, the bias term rapidly decreases as $M$ increases and nearly vanishes for $M \geq N$.

## J.2. Length Reward Experiments

In addition to the results with $N = 2, 4$ in Figure 3a, we provide the results with $N = 8, 16$ in Figure 5. We also show the distribution of the reference model, which is the initial model before training. We observe that the distribution of the reference model is approximately uniform and the baseline model produces only medicore length responses concentrated around 100 tokens. On the other hand, IAMA models generate diverse responses and as $N$ increases, the distribution is more polarized to produce both short and long responses. This aligns with the results in Section 3.1.

## J.3. HH-RLHF Experiments

We provide additional experimental results for HH-RLHF experiments.

**Experiments with $N = 8$** Figure 6 (right) shows the RLHF results with $N = 8$. Similar to the $N = 8$ case in Figure 3b, our method pushes the Pareto frontier significantly compared to the baseline methods.

**Experiments on Mistral 7B** We conduct additional experiments using Mistral 7B Instruct (Jiang et al., 2023), a state-of-the-art open-source LLM. We follow the same experimental setup as with Alpaca 7B except for $\beta = 0.01$ and $\mathrm{KL}_{\mathrm{target}} = 0.1$. Figure 7 shows the Pareto frontiers of IAMA and baseline with and without BoN sampling ($N = 4$). We see a similar tendency as Alpaca 7B results.

## J.4. Comparison with InfAlign

We compare IAMA with InfAlign (Balashankar et al., 2025) in the same setting as the RLHF experiments in Section 6.2. For InfAlign, we transform the calibrated rewards separately with $f(x) = e^{10x}$ (as suggested in the original paper) and

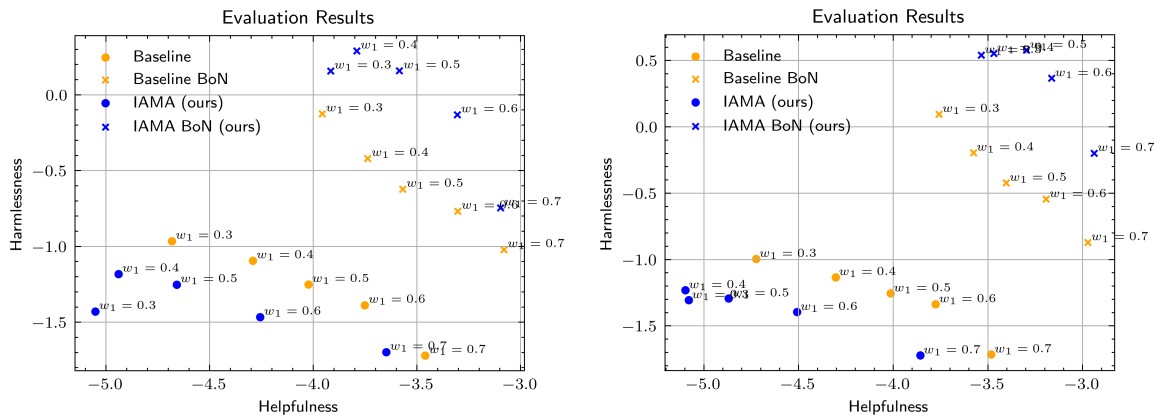

*Figure 6.* RLHF results with $N = 4$ (left) and $N = 8$ (right).

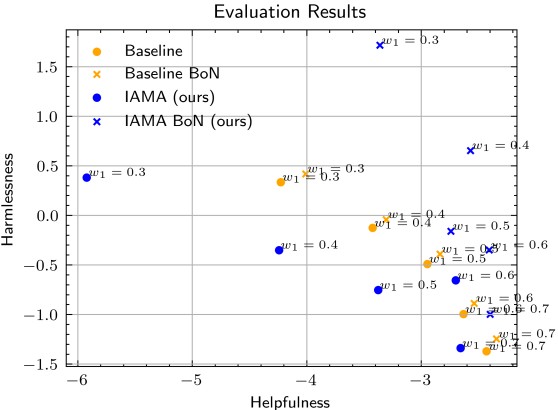

*Figure 7.* HH-RLHF results on Mistral 7B with $N = 4$.

average them with different weights. Table 3 shows the comparison of averaged BoN rewards evaluated by golden reward models. IAMA outperforms InfAlign across all weights, which may be because InfAlign is designed for a different setting (single reward and win-rate maximization).

*Table 3.* Comparison of averaged BoN rewards with InfAlign. The weight denotes the weight for helpfulness reward.

| Weight for helpfulness | 0.3 | 0.4 | 0.5 | 0.6 | 0.7 |
|---|---|---|---|---|---|
| IAMA (ours) | **8.94** | **10.22** | **11.44** | **12.72** | **13.98** |
| InfAlign | 8.73 | 9.94 | 11.27 | 12.49 | 13.83 |
| Baseline | 8.78 | 9.91 | 11.19 | 12.55 | 13.93 |

