# OpenReview forum: "Inference-Aware Meta-Alignment of LLMs via Non-Linear GRPO"
_ICML.cc/2026/Conference — ICML 2026 regular_

### Official Review · Reviewer_7D3Q · 2026-02-18

**Soundness:** 4
**Presentation:** 4
**Significance:** 3
**Originality:** 3
**Overall Recommendation:** 5
**Confidence:** 3

**Summary:**

*(I did not fully understand every derivation in the paper—especially the convergence analysis in Section 5—so I may miss some nuances in the theoretical part and defer to other reviewers on technical correctness.)*

This paper argues that modern LLM alignment is increasingly two-stage: training-time alignment is followed by inference-time alignment (e.g., Best-of-N=BoN, soft BoN), where users or systems select among multiple sampled responses. The authors claim that standard training objectives—typically optimizing a single expected reward or a weighted average of multiple rewards—tend to collapse the policy toward “average” outputs, reducing sample diversity and thereby limiting the gains from inference-time selection.

To formalize “training for inference-time alignment,” the paper proposes Inference-Aware Meta-Alignment (IAMA): optimize the expected reward after applying an inference-time transformation (e.g., BoN or soft BoN), potentially across multiple reward functions and an aggregation function g. However, this objective is non-linear in the policy, so standard TRPO/PPO/GRPO-style updates cannot be directly applied. The authors then introduce non-linear GRPO, which linearizes the non-linear functional objective via a first-order variation (functional derivative) around the current policy and uses the resulting pseudo-reward as a drop-in replacement in a GRPO-like proximal update. They derive concrete forms for BoN and soft BoN and provide efficient sample-based computation to implement the method.

Empirically, the paper shows that IAMA-trained policies preserve more “useful diversity” for downstream inference-time steering: for conflicting length preferences, the meta-aligned policy exhibits a more multi-modal length distribution than the baseline; for helpfulness vs harmlessness, applying BoN at inference time yields a better Pareto frontier for the IAMA model than the standard alignment baseline.

**Compliance With Llm Reviewing Policy:**

Affirmed.

**Final Justification:**

I fully understand the method's scope of contribution. 1) Definitely, it is novel and valuable for in-domain meta alignment, 2) but it is limited to pre-defined tasks only; it cannot be applied to ood domain. That is why I keep my positive score with slightly increased confidence. I think it is valuable paper for ICML audience.

**Key Questions For Authors:**

1. Looking at the IAMA training objective, it seems like the case where one sample ranks 1st out of N has a loss of 0, which appears to be the maximum. In that case, wouldn't simply applying normalization across the entire batch after computing the transformed reward already make training more efficient?

2. My current interpretation is that the main value is maximizing inference-time controllability for multi-objective alignment—i.e., learning a base proposal distribution that makes it easy to reweight/choose objectives at inference time—*rather than true meta-learning for arbitrary unseen reward models.* Do you agree with this framing? If so, which deployment scenarios do you consider most compelling (e.g., user-driven selection among a fixed set of criteria, dynamic policy via different T, changing mixture weights), and what evidence do you have for transfer to new mixtures or partially unseen objectives?

Even though the authors fail to give me a clear answer, I will keep my score. But, if succeed, I will increase my score.

**Limitations:**

IAMA only do meta-learn about *already seen reward*, cannot easily adapt to unseen arbitrary reward. It means it is hard to directly apply for real world meta learning.

**Strengths And Weaknesses:**

**Strengths**

- **Strong and very important motivation.** The paper targets a practical mismatch: training objectives are often optimized in isolation, even though deployment commonly relies on inference-time selection/steering. The IAMA formulation directly addresses this mismatch.

- **Clean conceptual framing.** Casting inference-time alignment as a policy transformation T[\pi] makes the “post-selection reward” objective explicit and general.

- **Practical algorithmic contribution.** The proposed non-linear GRPO can be implemented largely as a drop-in change (replace reward with an estimated functional derivative) while retaining GRPO/PPO-style tooling and stabilizers.

- **Empirical evidence supports the claim.** The experiments clearly illustrate that meta-aligned policies can better support inference-time steering under conflicting objectives, and can push the helpfulness/harmlessness Pareto frontier when BoN is used at inference.

**Weaknesses (I don't feel it is big weakness)**

- **Requires specifying the objective family in advance.** The method assumes access to a set of reward functions (and/or inference-time transformations) during training. In real-world personalization/adaptation settings, the target preference may be partially unknown or difficult to proxy; the method’s robustness to misspecified or incomplete reward sets has not been fully explored.

- **The scope of “meta” generalization could be clarified.** The framework looks most directly applicable to improving controllability under a known family of objectives and selection rules, rather than guaranteeing generalization to arbitrary unseen reward models.

---

> ### Author Rebuttal · Authors · 2026-03-26
>
> Thank you for your helpful feedback and positive comments on our contributions.
> We address the main concerns below.
>
> **"Requires specifying the objective family in advance"**
>
> We agree that our method assumes access to a set of reward functions during training. On the other hand, our formulation leads to diversity-seeking behavior, which may allow adaptation to new tasks (i.e., new rewards) via inference-time alignment.
>
> **"The scope of “meta” generalization could be clarified"**,
> **"My current interpretation is that the main value is maximizing inference-time controllability for multi-objective alignment"**
>
> Thanks for raising this point.
> While our formulation focuses on finite sets of tasks (rewards) for simplicity, we can generalize the situation to a distribution over tasks as in MAML. Pursuing this direction would be an interesting venue for future work. We will clarify this point in the final version of the paper.
>
> **"Looking at the IAMA training objective, it seems like the case where one sample ranks 1st out of N has a loss of 0, which appears to be the maximum. In that case, wouldn't simply applying normalization across the entire batch after computing the transformed reward already make training more efficient?"**
>
> Sample rewards (i.e. empirical functional derivatives) have a degree of freedom up to an additive constant shift. Therefore, the maximum value does not need to be zero; setting it to zero in this case has no particular significance.

---

> > ### Author Rebuttal · Reviewer_7D3Q · 2026-03-31
> >
> > Thanks for the clarification. I fully understand the method's scope of contribution. 1) Definitely, it is novel and valuable for in-domain meta alignment, 2) but it is limited to pre-defined tasks only; it cannot be applied to ood domain. That is why I keep my positive score with slightly increased confidence.

---

### Official Review · Reviewer_Aca3 · 2026-02-20

**Soundness:** 3
**Presentation:** 3
**Significance:** 3
**Originality:** 3
**Overall Recommendation:** 5
**Confidence:** 3

**Summary:**

This paper proposes Inference-Aware Meta-Alignment (IAMA), a two-stage alignment framework that meta-trains a base LLM so it can be efficiently adapted at inference time via algorithms such as Best-of-N (BoN) and Soft-BoN. The research assesses a significant question: how to align LLMs to multiple potentially conflicting reward functions under limited inference-time compute. The authors attempt to address a central concept in modern alignment: the interaction between training-time optimization and inference-time selection, by formulating the problem as a non-linear functional optimization in the space of probability measures.

To solve this, they introduce non-linear GRPO, a mirror-descent-style extension of GRPO that uses first-order variations of a non-linear reward functional. The paper provides convexity results for BoN-type objectives and proves linear convergence under KL-relative smoothness assumptions. Empirical results (length reward and helpfulness/harmlessness RLHF) show improved Pareto trade-offs when BoN is applied at inference.

**Compliance With Llm Reviewing Policy:**

Affirmed.

**Key Questions For Authors:**

What is the compute trade-off? Can you quantify total system cost (training + inference) relative to training separate aligned models?

**Limitations:**

Yes

**Strengths And Weaknesses:**

Strengths
- Clear Problem Motivation: The mismatch between training-time RLHF and inference-time BoN-style selection is well articulated and important.
- Conceptual Novelty: Framing inference-aware alignment as meta-learning in the space of probability measures is elegant and unifies prior inference-aware alignment work.
- Theoretical Contribution: Several theoretical contributions such as the formalization as non-linear optimization over distributions, convexity and KL-relative smoothness results for BoN-type objectives, and linear convergence guarantees
- Practical Simplicity: Implementation reduces to replacing the reward with an estimated functional derivative in GRPO.
- Empirical Evidence of Pareto Gains: Results show improved Pareto frontiers under inference-time BoN, demonstrating that meta-alignment produces diversity useful for selection.

Weaknesses
- Experiments are sort of narrow (length reward, helpfulness/harmlessness RLHF). Broader tasks (reasoning, safety stress tests, adversarial robustness) would strengthen claims of generality.
- Dependence on BoN structure- convexity and guarantees rely on BoN-type transformations. The method is claimed to be general, but theoretical backing outside BoN remains limited.
- The empirical gains appear moderate; the main novelty is theoretical rather than experimental.

---

> ### Author Rebuttal · Authors · 2026-03-26
>
> Thank you for your constructive feedback. We appreciate your positive comments on the theoretical contributions of our work. We address the main concerns below.
>
> **"Experiments are sort of narrow"**
>
> As the reviewers noted, this paper makes clear contributions in the novel formulation, algorithm proposal, and theoretical analysis. The experimental results sufficiently support the claims of the paper by demonstrating that this formulation provides a diverse-seeking policy, which allows effective adaptation to different reward functions at inference time,
> and such policies can be learned with non-linear GRPO.
>
> **"Dependence on BoN structure- convexity and guarantees rely on BoN-type transformations."**
>
> We agree that our theoretical guarantees rely on the convexity proven for BoN-type methods.
> However, BoN is widely used and an important use case.
> In addition, the algorithm and formulation remain valid even outside this setting.
>
> **"What is the compute trade-off? Can you quantify total system cost (training + inference) relative to training separate aligned models?"**
>
> Compared to our approach, training separate $n$ models would require $n$ times the training cost (if computing rewards is not the bottleneck) and deploying separate $n$ models would require $n$ times the memory cost. Considering the recent trend of increasing model sizes, this cost difference is significant. While our approach may introduce additional inference cost due to sampling-based alignment (e.g., BoN), it avoids the need to maintain and train multiple large models, leading to a more favorable overall system trade-off in practice.

---

> > ### Author Rebuttal · Reviewer_Aca3 · 2026-03-31
> >
> > My concerns are fully addressed, thanks

---

### Official Review · Reviewer_UXYe · 2026-03-12

**Soundness:** 2
**Presentation:** 3
**Significance:** 3
**Originality:** 3
**Overall Recommendation:** 4
**Confidence:** 3

**Summary:**

This paper proposes the inference-aware meta-alignment (IAMA) framework, which is a novel approach to train LLM to adapt to different user preferences via inference-time alignment algorithms like BoN sampling. And, as this introduces a non-linear optimization problem, they also proposed the non-linear GRPO that approximates the functional derivative of the reward objective using empirical samples with provable convergence guarantee to the optimal solution.

**Compliance With Llm Reviewing Policy:**

Affirmed.

**Final Justification:**

The rebuttal addressed my concerns. The authors added extra experiments to support the theoretical finding, which makes their claim stronger.

**Key Questions For Authors:**

- Would you be able to provide more evaluation on existing benchmarks?
- Would you be able to provide comparison towards more baselines?

**Limitations:**

Yes. Note also IAMA seems to heavily rely on BoN sampling. Without BoN sampling, the models may not outperform standard aligned models.

**Strengths And Weaknesses:**

Strength:

- Very strong theoretical foundations. The proofs illustrate the convexity and smoothness of BoN-type objectives and thus provide  theoretical provable convergence guarantee to the framework.
- The non-linear GRPO seems to pretty practical and useful as an easier-to-optimize alternative for GRPO, and it seems that it can also extend to other non-linear RLHF objectives.

Weakness:

- The experimental setting seems to be very limited. The method is only evaluated on one one length-control task and one standard hh-rlhf task. More explicit testing and comparison on existing benchmarks (for example, GSM8K, MT-Bench, AlpacaEval) might be needed to further show the effectiveness of the method.
- The paper only compares to standard GRPO as their baseline. The related work mentions like InfAlign is relavant but does not empirically compare the performance (I understand the practical focuses are different. But since they tackle on similar goals, I still think they are highly relevant to compare)
- The paper uses Qwen 32B as their golden rewards to evaluate the aligned models. But to my knowledge, models as GPT-4 ish, Claude, or bigger Llamma models are used more often.

---

> ### Author Rebuttal · Authors · 2026-03-26
>
> Thank you for your constructive feedback and for highlighting the strengths of our theoretical contributions. We address the main concern below.
>
> **"The experimental setting seems to be very limited."**
>
> Thank you for raising this point.
> As the reviewers noted, this paper makes clear contributions in the novel formulation, algorithm proposal, and theoretical analysis. The experimental results sufficiently support the claims of the paper by demonstrating that this formulation provides a diverse-seeking policy, which allows effective adaptation to different reward functions at inference time,
> and such policies can be learned with non-linear GRPO.
>
> In addition, InfAlign is designed for a different setting (as the reviewer pointed out), and it requires generating multiple rollouts for each prompt with the base model in advance, which is computationally expensive and not compatible with standard RLHF training pipelines.
>
> That being said, we agree that comparison with other baselines including InfAlign would be interesting. We show the comparison of averaged BoN rewards with InfAlign in the table below. We follow the same setting as in our RLHF expriments in section 6.2. For InfAlign, we transform the calibrated rewards seperately with $\phi(x) = e^{10 x}$ (suggested in the original paper) and average them with different weights.
> We see that IAMA outperforms InfAlign across all weights, which may be because InfAlign is designed for a different setting (single reward and win-rate maximization).
> We will include this comparison in the final version of the paper.
>
> | weight for helpfulness           |     0.3 |     0.4 |     0.5 |     0.6 |     0.7 |
> |----------------|--------|--------|--------|--------|--------|
> | IAMA (ours) |  **8.94** | **10.22** | **11.44** | **12.72** | **13.98** |
> | InfAlign  |  8.73 |  9.94 | 11.27 | 12.49 | 13.83 |
> | Baseline   |  8.78 |  9.91 | 11.19 | 12.55 | 13.93 |

---

> > ### Author Rebuttal · Reviewer_UXYe · 2026-04-01
> >
> > Thanks for the authors response. My concerns are fully addressed. I'll update my score accordingly.

---

### Official Review · Reviewer_HbkZ · 2026-03-13

**Soundness:** 4
**Presentation:** 4
**Significance:** 4
**Originality:** 4
**Overall Recommendation:** 5
**Confidence:** 3

**Summary:**

This paper proposes an inference-aware meta-alignment (IAMA) framework for aligning LLMs to diverse user preferences via two-stage alignment procedures. This paper formulates IAMA as a non-linear optimization problem and proposes the non-linear GRPO algorithm to solve the problem. The paper provides convergence guarantees of the proposed algorithm. Experimental results demonstrated the effectiveness of IAMA via non-linear GRPO in aligning LLMs to multiple reward functions simultaneously.

**Compliance With Llm Reviewing Policy:**

Affirmed.

**Key Questions For Authors:**

Could the authors provide additional experimental comparisons with other inference-aware baselines or alternative deployment strategies (e.g., multiple specialized models, reward-conditioned approaches, or prior inference-aware training methods in overlapping settings)?

**Limitations:**

yes

**Strengths And Weaknesses:**

Strengths:
1. This paper studies the problem of aligning LLMs to diverse user preferences via two-stage alignment procedures, and formulates it as a non-linear optimization problem. This problem is important and worth studying.
2. The paper provides convergence guarantees of the proposed algorithm, making the paper theoretically sound.
3. Experimental results demonstrated the effectiveness of IAMA via non-linear GRPO in aligning LLMs to multiple reward functions simultaneously.

Weaknesses:

The empirical evaluation is somewhat limited relative to the scope of the paper’s claims. While the experiments provide some validation of the theoretical motivation, they are not sufficient to fully demonstrate the practical impact of the proposed method. In particular, the primary baseline appears to be standard GRPO trained on an averaged reward. The paper does not include sufficient comparisons with other inference-aware baselines or alternative deployment strategies (e.g., multiple specialized models, reward-conditioned approaches, or prior inference-aware training methods in overlapping settings). As a result, it remains unclear whether the observed gains are due to the proposed framework itself or simply the use of a BoN-aware training objective. This weakens the empirical support for both the novelty and the practical significance of the method.

---

> ### Author Rebuttal · Authors · 2026-03-26
>
> Thank you for your helpful feedback.
> We appreciate your positive comments on the theoretical contributions of our work.
> We address the main concern below.
>
> **"The empirical evaluation is somewhat limited relative to the scope of the paper’s claims."**, **"Could the authors provide additional experimental comparisons with other inference-aware baselines or alternative deployment strategies"**
>
> Thank you for raising this point.
> As the reviewers noted, this paper makes clear contributions in the novel formulation, algorithm proposal, and theoretical analysis. The experimental results sufficiently support the claims of the paper by demonstrating that this formulation provides a diverse-seeking policy, which allows effective adaptation to different reward functions at inference time,
> and such policies can be learned with non-linear GRPO.
>
> That being said, we agree that comparison with other baselines including InfAlign would be interesting. We provide the comparison in the response to Reviewer UXYe. Please refer to the response for details. We will include this comparison in the final version of the paper.

---

> > ### Author Rebuttal · Reviewer_HbkZ · 2026-04-03
> >
> > Thanks for your response. I will maintain my positive score.

---

### Decision · Program_Chairs · 2026-04-30

**Decision:**

Accept (regular)

**Comment:**

This paper proposes a two-stage method for inference-time alignment (Inference-Aware Meta-Alignment; IAMA), which first meta-trains a base LLM to be efficiently adaptable at inference time via Best-of-N (BoN) and Soft-BoN. The goal is to align LLMs to multiple potentially conflicting reward functions under limited inference-time compute. The problem is formulated as a non-linear functional optimization in the space of probability measures (nonlinear GRPO).

All reviewers agree that this is a solid paper. They praised the clear motivation, novelty, and theoretical contribution of this paper. The main criticism is the relatively narrow experiments and lack of comparison against strong baselines. This was mitigated in the rebuttal where a comparison against InfAlign was shown.

Overall, while the empirical validation is somewhat limited, the main contribution of this paper is mostly theoretical and I think it is of interest to the ICML community.